# A maternally programmed intergenerational mechanism enables male offspring to make piRNAs from Y-linked precursor RNAs in *Drosophila*

Zsolt G. Venkei[1,11], Ildar Gainetdinov ®[2,11], Ayca Bagci[2], Margaret R. Starostik ®[3], Charlotte P. Choi[3], Jaclyn M. Fingerhut ®[1,4], Peiwei Chen ®[5], Chiraag Balsara[6], Troy W. Whitfield ®[1], George W. Bell[1], Suhua Feng[7,8], Steven E. Jacobsen ®[7,8,9], Alexei A. Aravin ®[5], John K. Kim[3], Phillip D. Zamore ®[2,10] ✉ & Yukiko M. Yamashita ®[1,4] ✉

In animals, PIWI-interacting RNAs (piRNAs) direct PIWI proteins to silence complementary targets such as transposons. In *Drosophila* and other species with a maternally specified germline, piRNAs deposited in the egg initiate piRNA biogenesis in the progeny. However, Y chromosome loci cannot participate in such a chain of intergenerational inheritance. How then can the biogenesis of Y-linked piRNAs be initiated? Here, using *Suppressor of Stellate* (*Su(Ste)*), a Y-linked *Drosophila melanogaster* piRNA locus as a model, we show that *Su(Ste)* piRNAs are made in the early male germline via 5′-to-3′ phased piRNA biogenesis initiated by maternally deposited *1360/Hoppel* transposon piRNAs. Notably, deposition of *Su(Ste)* piRNAs from XXY mothers obviates the need for phased piRNA biogenesis in sons. Together, our study uncovers a developmentally programmed, intergenerational mechanism that allows fly mothers to protect their sons using a Y-linked piRNA locus.

In animals, the PIWI-interacting RNA (piRNA) pathway generates small RNAs that direct silencing of transposable elements and other selfish genetic elements[1]. Loss of piRNAs derepresses transposons[2–5], dysregulates gene expression[6–8] and reduces fertility. At the core of piRNA-mediated silencing are 18–35-nucleotide (nt) piRNAs that bind to and guide PIWI proteins to their targets via nucleotide sequence

complementarity[2,9–12]. The three *D. melanogaster* PIWI proteins have specialized functions in the germline: Piwi represses transposon transcription in the nucleus, whereas Ago3 and Aubergine (Aub) cleave piRNA precursor and transposon transcripts in the cytoplasm[4,12–22].

Animals often use pre-existing piRNAs to direct slicing of complementary transcripts and initiate piRNA biogenesis from the resulting

[1]Whitehead Institute for Biomedical Research, Department of Biology, Massachusetts Institute of Technology, Cambridge, MA, USA. [2]RNA Therapeutics Institute, University of Massachusetts Chan Medical School, Worcester, MA, USA. [3]Department of Biology, Johns Hopkins University, Baltimore, MD, USA. [4]Howard Hughes Medical Institute, Whitehead Institute, Massachusetts Institute of Technology, Cambridge, MA, USA. [5]Division of Biology and Biological Engineering, California Institute of Technology, Pasadena, CA, USA. [6]Life Sciences Institute, University of Michigan, Ann Arbor, MI, USA. [7]Department of Molecular, Cell and Developmental Biology, University of California, Los Angeles, CA, USA. [8]Eli and Edyth Broad Center of Regenerative Medicine and Stem Cell Research, University of California, Los Angeles, CA, USA. [9]Howard Hughes Medical Institute, University of California Los Angeles, Los Angeles, CA, USA. [10]Howard Hughes Medical Institute, University of Massachusetts Chan Medical School, Worcester, MA, USA. [11]These authors contributed equally: Zsolt G. Venkei, Ildar Gainetdinov. ✉e-mail: phillip.zamore@umassmed.edu; yukikomy@wi.mit.edu

5′-monophosphorylated cleavage products[23]. For example, in the *D. melanogaster* female germline, Ago3 and Aub are loaded with piRNAs derived from complementary transcripts (transposon messenger RNAs and piRNA precursors), and the 3′ cleavage product of Ago3 slicing is used to make antisense Aub-loaded piRNAs and vice versa. This positive feedback loop—the 'ping-pong' cycle—amplifies the transposon-targeting population of piRNAs[4,24]. The ping-pong pathway also initiates 5′-to-3′ fragmentation of the remainder of the cleavage product into tail-to-head, phased piRNAs loaded in Piwi[19,20,25,26]. Phased piRNA biogenesis requires the endonuclease Zucchini (Zuc; PLD6 in mammals) and the RNA helicase Armitage (Armi; MOV10L1 in mammals)[27–30]. The ping-pong pathway increases only piRNA abundance, whereas production of phased primary piRNAs adds sequence diversity to the piRNA[19] pool.

The ping-pong cycle requires pre-existing piRNAs to initiate the amplification process. In *D. melanogaster*, maternally deposited piRNAs serve this purpose, providing a pool of piRNAs that can initiate the ping-pong cycle[17,31–34]. For example, the inability of naïve mothers to provide P-element-derived piRNAs when mated with P-element-infested fathers causes derepression of selfish elements and sterility in their offspring, a phenomenon called hybrid dysgenesis[32,35–43].

*Stellate (Ste)* and *Suppressor of Stellate (Su(Ste))* in *D. melanogaster* provided the founding paradigm of piRNA-directed repression[44–48]. *Ste* is a repetitive gene whose unchecked expression results in Ste protein crystals, amyloid-like protein aggregates that cause male sterility via unknown mechanisms[49]. To ensure male fertility, *Ste* genes on the X chromosome are normally repressed by *Su(Ste)* piRNAs that are antisense to *Ste* and are produced from Y chromosome transcripts[12,50–52]. *Su(Ste)* locus comprises tandem repeats nearly identical (~90%) to *Ste*. *Ste* is the major silencing target of the piRNA pathway in the *D. melanogaster* male germline[7,51–55], requiring *armi*, *zuc*, *krimp*, *spn-E*, *vas*, *aub* and *ago3*, but not *piwi* or *rhino (rhi)*, suggesting that *Ste* repression is primarily dependent on cytoplasmic cleavage of the *Ste* mRNA[12,27,56–59]. Because *Su(Ste)* is encoded on the Y chromosome, fly mothers—which lack a Y chromosome—cannot provide their sons with *Su(Ste)* piRNAs to initiate biogenesis. How the male germline produces *Su(Ste)* piRNAs in the absence of maternally deposited *Su(Ste)* piRNAs is unknown.

In this Article, we describe the mechanism by which the male germline represses *Ste* in the absence of maternally deposited *Su(Ste)* piRNAs. We show that *Su(Ste)* piRNAs are produced by Armi- and Zuc-dependent phased piRNA biogenesis in male germline stem cells (GSCs) and early spermatogonia (SGs), days before expression of *Ste* target RNAs in spermatocytes. Phased biogenesis of *Su(Ste)* piRNAs in GSCs/SGs is critical to repress *Ste* later in spermatocytes and thus for male fertility. Our data show that males from XX mothers use maternally deposited *1360/Hoppel* piRNAs to cleave *Su(Ste)* precursors and initiate 5′-to-3′ phased biogenesis of *Su(Ste)* piRNAs in the early germline (GSCs/SGs). We show that the requirement for Armi, a protein essential for phased piRNA biogenesis, in *Su(Ste)* piRNA production in males is relieved when XXY females provide maternal *Su(Ste)* piRNAs to their sons' germline. These data explain how maternally deposited piRNAs can direct production of non-homologous piRNA guides in the germline of the progeny. Our study reveals a mechanism for

intergenerational transmission of piRNA-coded memory in the absence of direct homology and demonstrates that the phased piRNA pathway can protect offspring from selfish genetic elements not encountered by their mothers.

## Results

### *Su(Ste)* transcription starts days before *Ste* expression

To investigate *Su(Ste)* piRNA precursor expression and processing into piRNAs during *D. melanogaster* spermatogenesis, we used single-molecule RNA fluorescent in situ hybridization (smRNA-FISH)[60,61]. By leveraging nucleotide polymorphisms between *Ste* and *Su(Ste)*, we used a single in situ probe to detect *Su(Ste)* and a collection of Stellaris in situ probes to visualize *Ste* (Methods). smRNA-FISH can detect *Ste* mRNA and *Su(Ste)* precursor transcripts but not mature piRNAs, because small RNAs are not retained in formaldehyde-fixed tissues[62] (Methods).

In wild-type testes, *Ste* transcripts were first detected in the nuclei of spermatocytes (Fig. 1a,b). In contrast, in XO males, which lack *Su(Ste)*, *Ste* transcripts were readily detected in the spermatocyte cytoplasm (Fig. 1c), leading to production of Ste protein crystals, a known cause of subfertility. Notably, in XO males, cytoplasmic *Ste* mRNA was observed only in spermatocytes (Fig. 1c), suggesting that *Ste* is transcriptionally silent in early germ cells (that is, GSCs and SGs).

Our smRNA-FISH experiments readily detected *Su(Ste)* expression in GSCs (Fig. 1b), earlier than previously reported[50]. Thus, *Su(Ste)* expression precedes that of *Ste* by ~2–3 days (Fig. 1a). In GSCs and SGs, *Su(Ste)* transcription was detectable only from the genomic strand that produces piRNA precursors antisense to *Ste* mRNA (Extended Data Fig. 1a). The steady-state abundance of nuclear antisense *Su(Ste)* transcripts peaked in late SGs/early spermatocytes and was undetectable by the time *Ste* expression was first detected, in late spermatocytes (Fig. 1b).

Ping-pong amplification of *Ste*-targeting piRNAs should require the presence of both antisense *Su(Ste)* and sense *Ste* RNA in the same cells. Our data, however, show that antisense *Su(Ste)* piRNA precursors are transcribed and processed into *Ste*-targeting piRNAs before the first detectable accumulation of *Ste* mRNA. Supporting the idea that antisense *Su(Ste)* precursors and sense *Ste* mRNA are not present in the same germ cell types, we did not detect short interfering RNAs (siRNAs) production from *Su(Ste)* loci (Fig. 1d). (siRNAs are produced by Dicer proteins from double-stranded RNAs[63]). We conclude that ping-pong amplification is unlikely to explain the biogenesis of *Su(Ste)* piRNAs in GSCs and SGs (Extended Data Fig. 1b,c).

### *Su(Ste)* transcripts are processed in early male germ cells

Consistent with earlier studies[27,54], we found that processing of antisense *Su(Ste)* precursors into mature piRNAs in GSCs/SGs depends on components of the phased piRNA biogenesis pathway. In wild-type GSCs/SGs, *Su(Ste)* transcripts were detected as a single nuclear focus, corresponding to nascent transcripts from the *Su(Ste)* loci (Fig. 2a). In contrast, in *armi*^1/72.1 or *zuc*^EY11457/− loss-of-function mutants, the nuclear foci of *Su(Ste)* transcripts were enlarged, and multiple cytoplasmic foci appeared, probably representing accumulation of unprocessed

**Fig. 1 | *Su(Ste)* transcription precedes that of *Ste* during germ cell differentiation. a**, Early stages of *D. melanogaster* spermatogenesis. The stem cell niche is formed by the non-dividing somatic cells of the hub (asterisk). The GSCs are physically attached to the hub and divide asymmetrically. The gonialblasts (GBs), the differentiating daughters of GSCs, undergo four rounds of mitotic divisions with incomplete cytokinesis. Resultant 16-cell SGs then enter meiotic prophase as spermatocytes. The expression patterns of *nos-gal4* and *bam-gal4* drivers in the adult male germ line are also indicated. GSCs and early SGs are indicated by a yellow dotted line; cyan lines indicate the zone of spermatocytes in this and all subsequent figures. **b**(i), **c**(i) Expression of *Ste* mRNA (red) and antisense *Su(Ste)* precursor (green) in the wild-type (**b**) and in

XO (**c**) testes (smRNA-FISH). Magnified view of boxed areas is shown in **b**(ii), **b**(iii), **c**(ii) and **c**(iii). Arrow points to *Su(Ste)* transcripts in a GSC nucleus. **b** and **c** represent *z*-projections that cover the depth of the testes, whereas **b**(ii), **b**(iii) **c**(ii) and **c**(iii) only cover the depth of the cells presented. Dotted white lines indicate the nuclear periphery. Red, *Ste* RNA; green, antisense *Su(Ste)* RNA; blue, DAPI. Scale bars, 20 μm (**b**(i), **c**(i)) and 5 μm (**b**(ii), **b**(iii), **c**(ii) and **c**(iii)). **d**, Length profile of *Ste*-, *Su(Ste)*- (Supplementary Table 3) and *flamenco*-derived small RNAs in control (*y¹w¹¹¹⁸/Y; nos-gal4:VP16/TM2*) testis. *flamenco* produces 21-nt siRNAs[79]. The data are the mean from two independent biological samples. Source numerical data are available in source data.

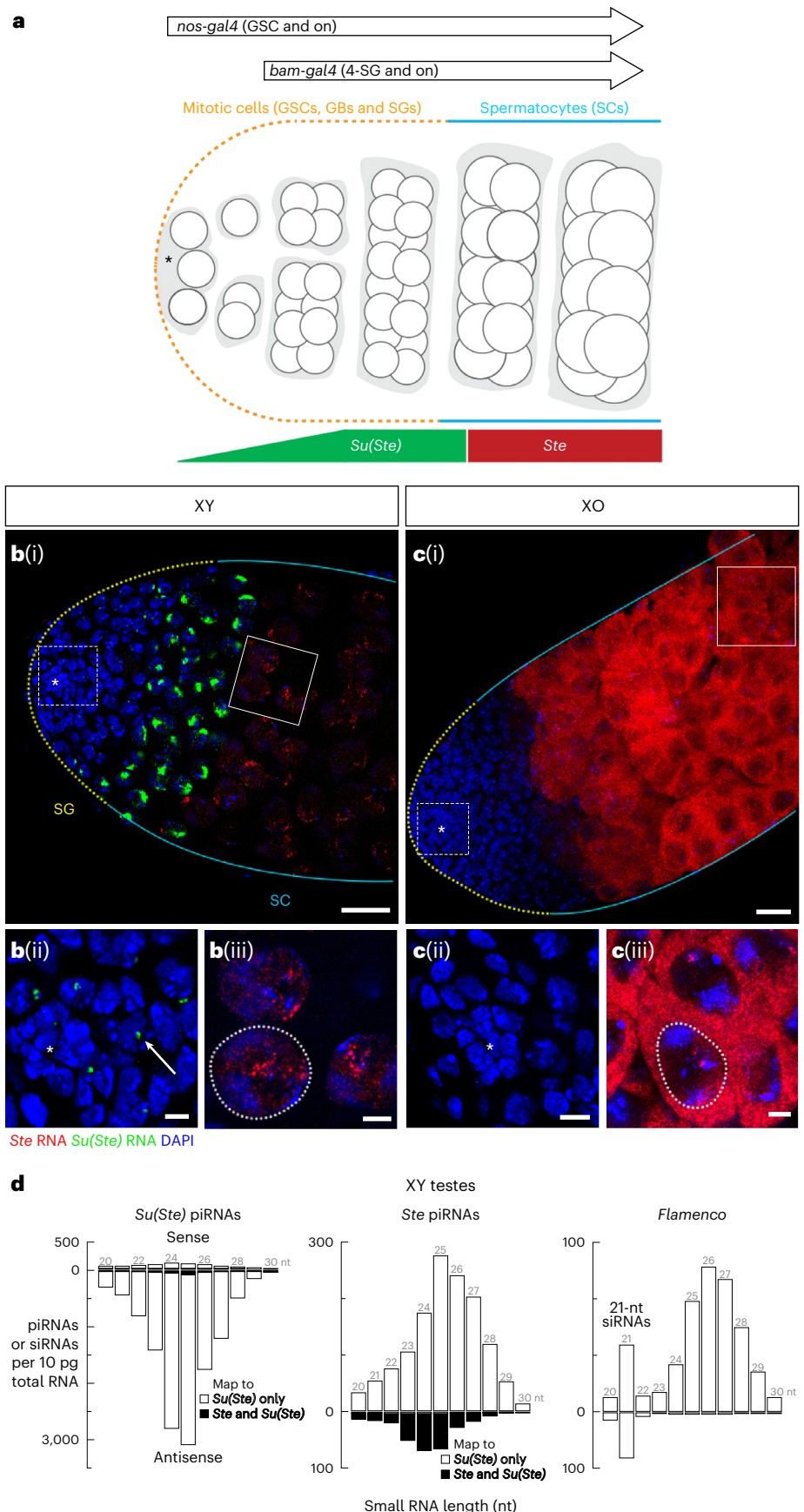

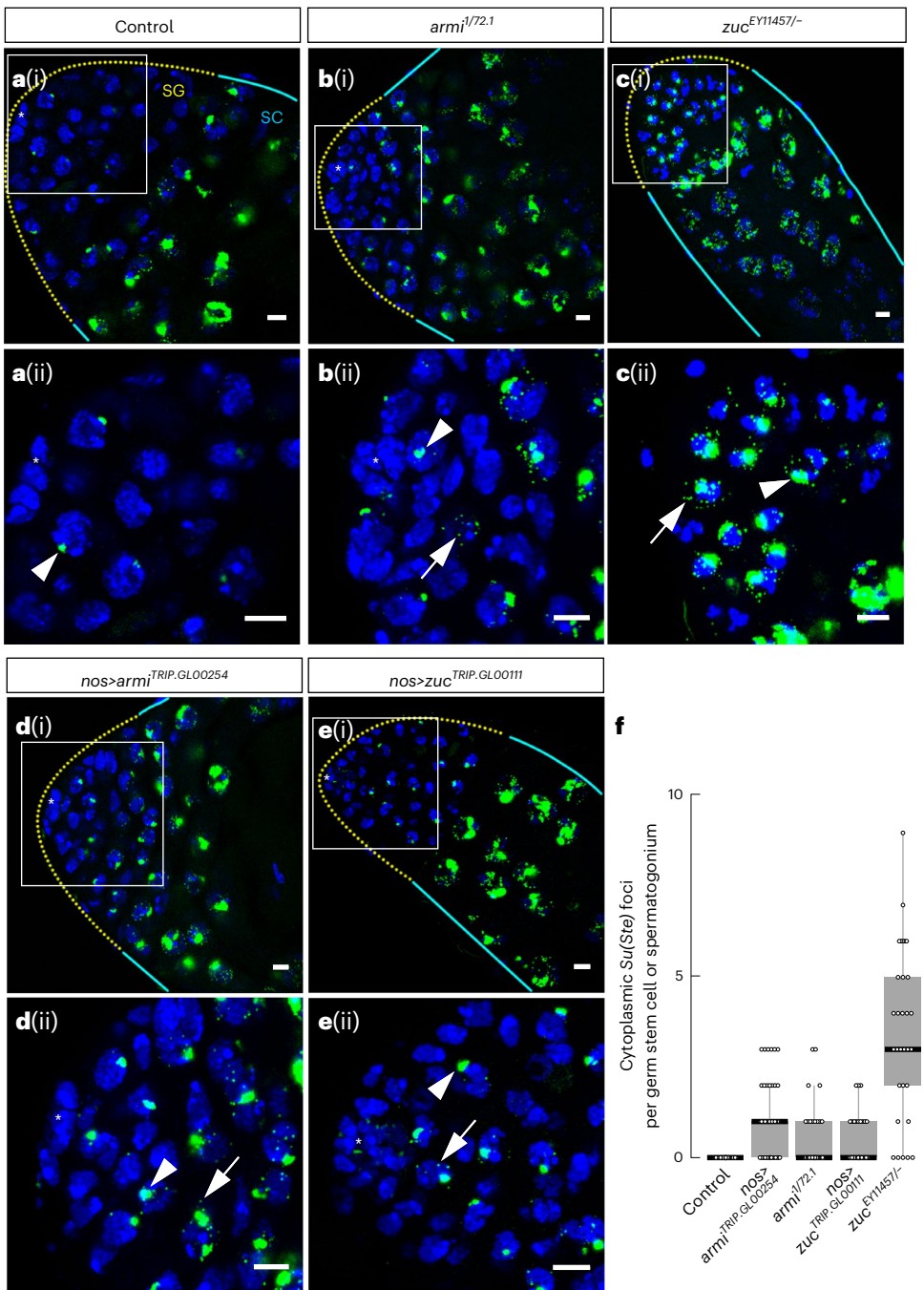

**Fig. 2 | *Su(Ste)* precursor transcripts accumulate in GSCs and SGs of *armi* and *zuc* mutant testes. a–e**, smRNA-FISH for antisense *Su(Ste)* precursor transcript (green) in control *y¹w¹¹¹⁸* testis (**a**) and in piRNA pathway mutant testes of the indicated genotypes: *armi* mutant (**b**); *zuc* mutant (**c**); *armi* RNAi (**d**); *zuc* RNAi (**e**)). The corresponding magnified regions of the niche marked by quadrates are shown in **a**(ii), **b**(ii), **c**(ii), **d**(ii) and **e**(ii) GSC and early SGs are indicated by yellow dotted lines; cyan lines indicate zone of spermatocytes. Arrowheads point to nuclear transcripts; arrows point to cytoplasmic RNA foci. The asterisks indicate the hub. Blue, DAPI. Scale bars, 5 μm. **f**, Quantification of cytoplasmic *Su(Ste)* RNA foci in GCSs and SG cells. Signal intensity was measured by maximum projection of *z*-stacks that encompass the entire cell. Box plots show the median and interquartile range (IQR); whiskers denote 1.5× IQR (*n* = 90 for control; *n* = 54 for *nos>armi^{TRIP.GL00254}*; *n* = 33 for *armi^{1/72.1}*; *n* = 34 for *nos>zuc^{TRIP.GL00111}*; *n* = 31 for *zuc^{EY11457/−}*). *P* = 2.2 × 10⁻¹⁶ for Kruskal–Wallis test (one-way analysis of variance on ranks) comparing all genotypes and control; Benjamini–Hochberg-corrected *P* values for post hoc pairwise two-tailed Mann–Whitney tests: *P* = 2 × 10⁻¹⁶ for *nos>armi^{TRIP.GL00254}* versus control; *P* = 7.2 × 10⁻⁹ for *armi^{1/72.1}* versus control; *P* = 9.4 × 10⁻⁹ for *nos>zuc^{TRIP.GL00111}* versus control; *P* = 2 × 10⁻¹⁶ for *zuc^{EY11457/−}* versus control. Source numerical data are available in source data.

piRNA precursor transcripts (Fig. 2b,c). Similar *Su(Ste)* cytoplasmic foci were detected when *armi* or *zuc* mRNA was specifically depleted in germ cells by RNA interference (RNAi) using pVALIUM22 transgenes (*armi^{TRIP.GL00254}* and *zuc^{TRIP.GL00111}*; henceforth, *armi^{RNAi}* and *zuc^{RNAi}*) driven by *nanos*(*nos*)-*Gal4* (ref. 64; Figs. 1a and 2d,e). The appearance of *Su(Ste)*

cytoplasmic foci in *zuc* and *armi* mutants (Fig. 2f) concurs with the increase in the steady-state abundance of *Su(Ste)* transcripts measured by quantitative reverse transcription polymerase chain reaction (qRT–PCR) in *zuc^{EY11457/−}* mutant testis enriched for SGs by over-expressing *dpp*: *Su(Ste)* precursors increased 1.9 ± 0.7-fold in mutants versus control

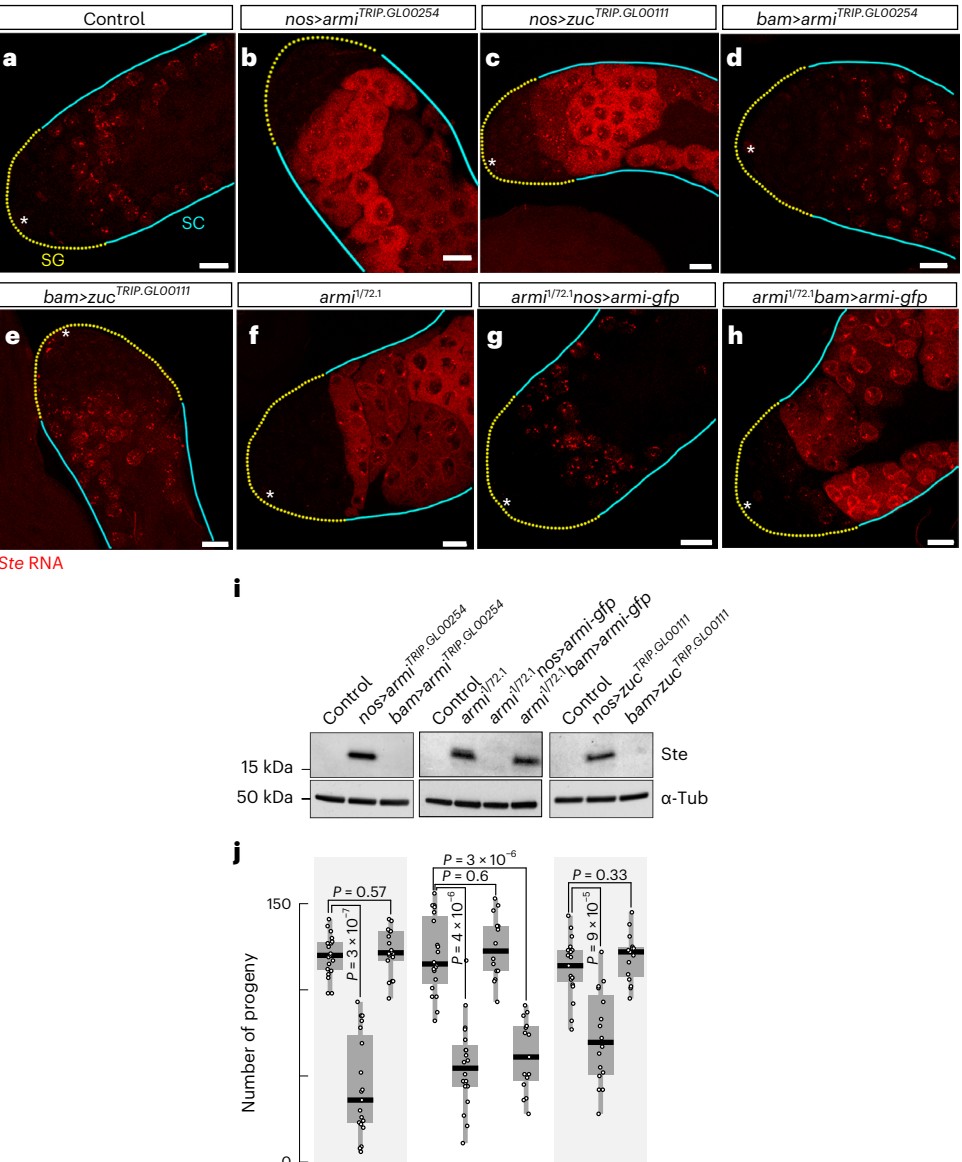

**Fig. 3 | *armi* and *zuc* are required in GSCs and early SGs to repress *Ste*.**
**a**–**h**, *Ste* smRNA-FISH (red) in the testes in control *y¹w¹¹¹⁸* testis (**a**) and in piRNA
pathway mutant testes of the indicated genotypes: *armi nos*-driven RNAi (**b**);
*zuc nos*-driven RNAi (**c**); *armi bam*-driven RNAi (**d**); *zuc bam*-driven RNAi (**e**);
*armi* mutant (**f**); *armi* mutant, *nos* rescue (**g**); *armi* mutant, *bam* rescue (**h**). GSCs
and early SGs are indicated by yellow dotted lines; cyan lines indicate zone of
spermatocytes. The asterisks indicate the hub. Scale bars, 20 μm. **i**, Anti-Ste and

anti-Tubulin western blots of whole testis lysates from the indicated genotypes.
**j**, Male fertility of indicated genotypes (number of progeny/male/7 days). Box
plots show the median and IQR; whiskers denote 1.5× IQR (*n* = 20 males per
genotype). *P* < 10⁻⁵ for Kruskal–Wallis test (one-way analysis of variance on ranks)
comparing all genotypes and controls; Benjamini–Hochberg-corrected *P* values
for all post hoc pairwise two-tailed Mann–Whitney tests are shown. Source
numerical data and unprocessed blots are available in source data.

testis (two-tailed, one sample *t*-test, *P* = 0.025), while *act5C* transcripts
changed 1.1 ± 0.7-fold (two-tailed, one sample *t*-test, *P* = 0.7; Extended
Data Fig. 2 and Supplementary Table 1).

By contrast, *Su(Ste)* piRNA precursor transcripts did not accumu-
late when *vas*—the helicase required for ping-pong piRNA process-
ing[19,65]—was depleted by *nos*-driven RNAi (Extended Data Fig. 3a–d).
Similarly, depletion of either of the endonucleases in the ping-pong
pathway (Aub or Ago3) did not stabilize *Su(Ste)* precursor transcripts
in GSCs/SGs (Extended Data Fig. 3a–d).

In the phased piRNA biogenesis pathway, the endonuclease Zuc
fragments piRNA precursors into head-to-tail pre-piRNAs, and the
overwhelming majority of phased pre-piRNAs bear a uridine as their
5′-terminal nucleotide (pre-piRNAs become mature piRNAs after their
3′ ends are trimmed and 2′-*O*-methylated). Conversely, piRNA guides

produced by the ping-pong pathway frequently have an adenine at
position 10, because endonucleases in the ping-pong pathway often
have an intrinsic preference for targets with an adenine at the posi-
tion that then becomes the tenth nucleotide of a new mature piRNA[66].
Transposon-derived piRNAs in testis are made by both the ping-pong
and the phased biogenesis pathways[54], and thus exhibit both the enrich-
ment of uridines as the first nucleotide (67 ± 3%) and a higher fre-
quency of adenines as the tenth nucleotide (37.2 ± 0.3%; Extended Data
Fig. 3e). Supporting the idea that processing of *Su(Ste)* precursors into
piRNAs in GSCs/SGs is catalysed by Zuc[19,20], we find that, although the
majority of *Su(Ste)*-derived piRNAs begin with a uridine (77 ± 1% at posi-
tion 1 versus 28.4 ± 0.2% at all positions), they show no enrichment for
adenine as the tenth nucleotide (21 ± 1% at position 10 versus 25.9 ± 0.3%
at all positions; Extended Data Fig. 3e). Together, these results suggest

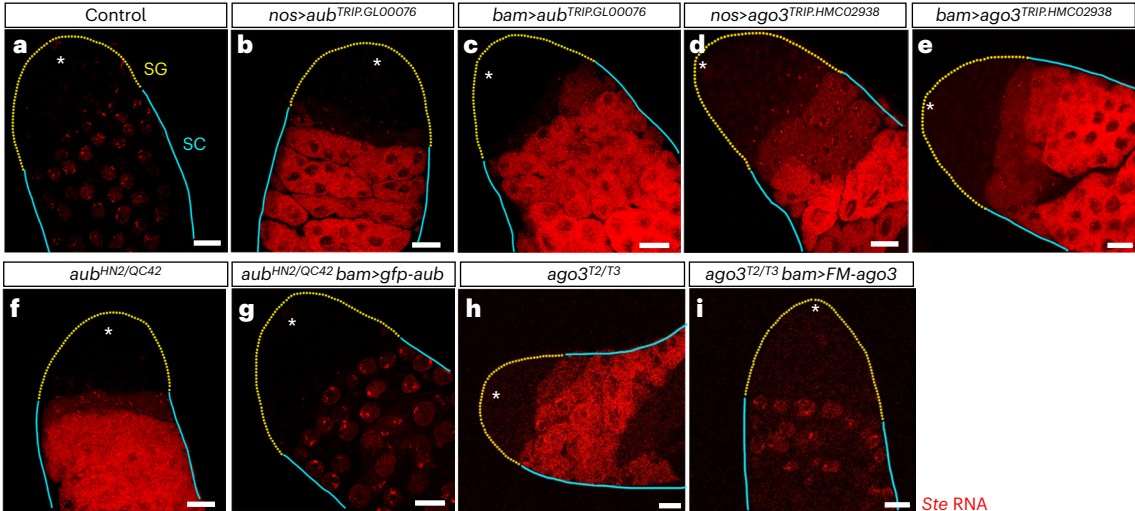

**Fig. 4 | To repress *Ste*, *aub* and *ago3* are required no later than the spermatogonial four-cell stage. a–i**, Representative images of *Ste* smRNA-FISH (red) in the testes in control *y¹w^IIIS* testis (**a**) and in piRNA pathway mutant testes of the indicated genotypes: *aub nos*-driven RNAi (**b**); *aub bam*-drive RNAi (**c**); *ago3 nos*-driven RNAi (**d**); *ago3 bam*-driven RNAi (**e**); *aub* mutant (**f**); *aub* mutant, *bam*-driven rescue (**g**); *ago3* mutant (**h**), *ago3* bam-driven rescue (**i**). GSCs and early SGs are indicated by a yellow dotted line; cyan lines indicate spermatocytes. The asterisks indicate the hub. Scale bars, 20 μm. Experiments were repeated three times with similar results. These results show that Aub and Ago3

programmed with antisense *Su(Ste)* piRNAs are required for efficient repression of *Ste*. Note that, in fly testes and ovaries, transposon-derived piRNAs partition between Aub and Ago3: most antisense, phased, 1U-enriched piRNAs are bound to Aub, while most sense, ping-pong produced, 10A-biased piRNAs are loaded in Ago3 (refs. 4,54). Yet antisense, phased, 1U-enriched *Su(Ste)* piRNA are loaded into both Aug and Ago3 (ref. 54). Our analyses also show that piRNAs produced from the cleavage products of slicing of *Ste* transcripts by *Su(Ste)* piRNAs (that is, responder *Ste* piRNAs[1]) are most frequently loaded in Ago3 (>51 ± 8% in Ago3 versus >7 ± 2% in Aub).

that, in GSC/SGs, the phased piRNA biogenesis pathway dominates the production of piRNAs from *Su(Ste)* transcripts.

### *Ste* silencing requires *zuc* and *armi* in early male germ cells

Repression of *Ste* in late spermatocytes depends on *zuc* and *armi* expression during a short window in early spermatogenesis. When *armi* or *zuc* mRNA was depleted by *nos*-driven RNAi (*nos>armi^RNAi* or *nos>zuc^RNAi*) throughout the germline (Fig. 1a), we observed derepression of *Ste* RNA (Fig. 3a–c), Ste protein accumulation (Fig. 3i) and reduced fertility (Fig. 3j). In contrast, using *bam-gal4* (Fig. 1a) to deplete *armi* or *zuc* in >4-cell SG stages (*bam>armi^RNAi* or *bam>zuc^RNAi*) had no observable effect on *Ste* repression or fertility (Fig. 3d,e), suggesting that *armi* and *zuc* are dispensable for *Ste* repression after the four-cell spermatogonial stage.

Consistent with the idea that *Ste* silencing requires Armitage in early germ cells, expression of an *armi-gfp* transgene under the control of *nos-gal4* restored *Ste* repression in *armi^1/72.1* testes (Fig. 3f,g). In contrast, expression of the same rescue construct driven by *bam-gal4* failed to rescue the *armi* mutant phenotype (Fig. 3h). Collectively, these data suggest that *Su(Ste)* piRNAs are produced in early germ cells by the phased biogenesis pathway.

### *Ste* silencing requires both Aub and Ago3

In the phased biogenesis pathway, the products of Zuc-catalysed fragmentation of piRNA precursors are loaded into PIWI Argonaute proteins and mature to become piRNAs[20,23]. In fly testis, >80% of *Su(Ste)*-derived piRNAs in Aub and Ago3 are derived from the antisense precursor transcript[54], suggesting that both proteins are programmed with antisense *Su(Ste)* piRNAs during phased biogenesis in GSC/SGs. Both Aub and Ago3 are required for repression of *Ste* mRNA in spermatocytes[54] (Fig. 4a–e). Antisense *Su(Ste)*-piRNA-guided Aub and Ago3 are thus non-redundant in silencing *Ste*.

We find that efficient repression of *Stellate* occurs when expression of Aub and Ago3 begins no later than the spermatogonial four-cell stage, that is, before *Su(Ste)* precursor transcription reaches its peak (Fig. 1a,b). Expressing a *gfp-aub* rescue transgene using *bam-gal4*

driver restored *Ste* repression in loss-of-function *aub^HN2/QC42* mutants (Fig. 4f,g). *Ste* was also silenced when a *bam*-driven *FLAG-Myc-ago3* rescue transgene was expressed in *ago3^T2/T3* mutant males (Fig. 4h,i). We conclude that both Aub and Ago3 programmed with antisense *Su(Ste)* piRNAs are required for efficient repression of *Ste*.

### *1360* piRNAs trigger phased biogenesis of *Su(Ste)* piRNAs

Efficient repression of *Ste* requires production of *Su(Ste)* piRNAs days before *Ste* is first expressed (Fig. 1). Production of *Su(Ste)* piRNAs in early male germ cells requires Zuc and Armi, components of the phased piRNA biogenesis pathway (Figs. 2 and 3). Typically, phased piRNA biogenesis is initiated by a PIWI protein-catalysed, piRNA-directed slicing event that generates a long 5′-monophosphorylated 3′-cleavage product (pre-pre-piRNA). The pre-pre-piRNA is then fragmented by Zuc into phased, tail-to-head pre-piRNAs[19,20,25,66]. But *Ste* piRNAs that could trigger phased fragmentation of *Su(Ste)* precursors are not produced by mothers (see below).

We propose that maternally inherited *1360/Hoppel* transposon-derived piRNAs initiate phased production of *Su(Ste)* piRNAs that direct cleavage of the *1360/Hoppel* sequence residing at the 5′ end of *Su(Ste)* precursor RNAs (Fig. 5a). Several observations support this idea: (1) transcription of *Su(Ste)* starts inside a *1360/Hoppel* transposon insertion upstream of the sequence complementary to *Ste* (ref. 52); (2) ovaries contain abundant *1360/Hoppel* transposon-derived piRNAs (~18,200 ± 400 per 10 pg total RNA); and (3) mothers deliver *1360/Hoppel* piRNA to their male offspring via the oocyte[32].

To test this model, we sequenced ≥ 200-nt long, 5′-monophosphorylated RNAs from adult testis to identify putative pre-pre-piRNAs. Like all Argonautes, PIWI proteins cleave their targets between nucleotides t10 and t11, the target nucleotides complementary to piRNA nucleotides g10 and g11. In the piRNA producing loci *42AB* and *petrel*, the 5′ ends of long RNAs most frequently lay between nucleotides g10 and g11 of an antisense piRNA, supporting the idea that these monophosphorylated RNAs are bona fide pre-pre-piRNAs ($Z_{10}$ = 5.1, $P = 6 × 10^{-7}$ for *42AB*; $Z_{10}$ = 8.5, $P = 2.3 × 10^{-17}$ for *petrel*; Extended Data Fig. 4a). As expected, we detected no antisense piRNAs overlapping

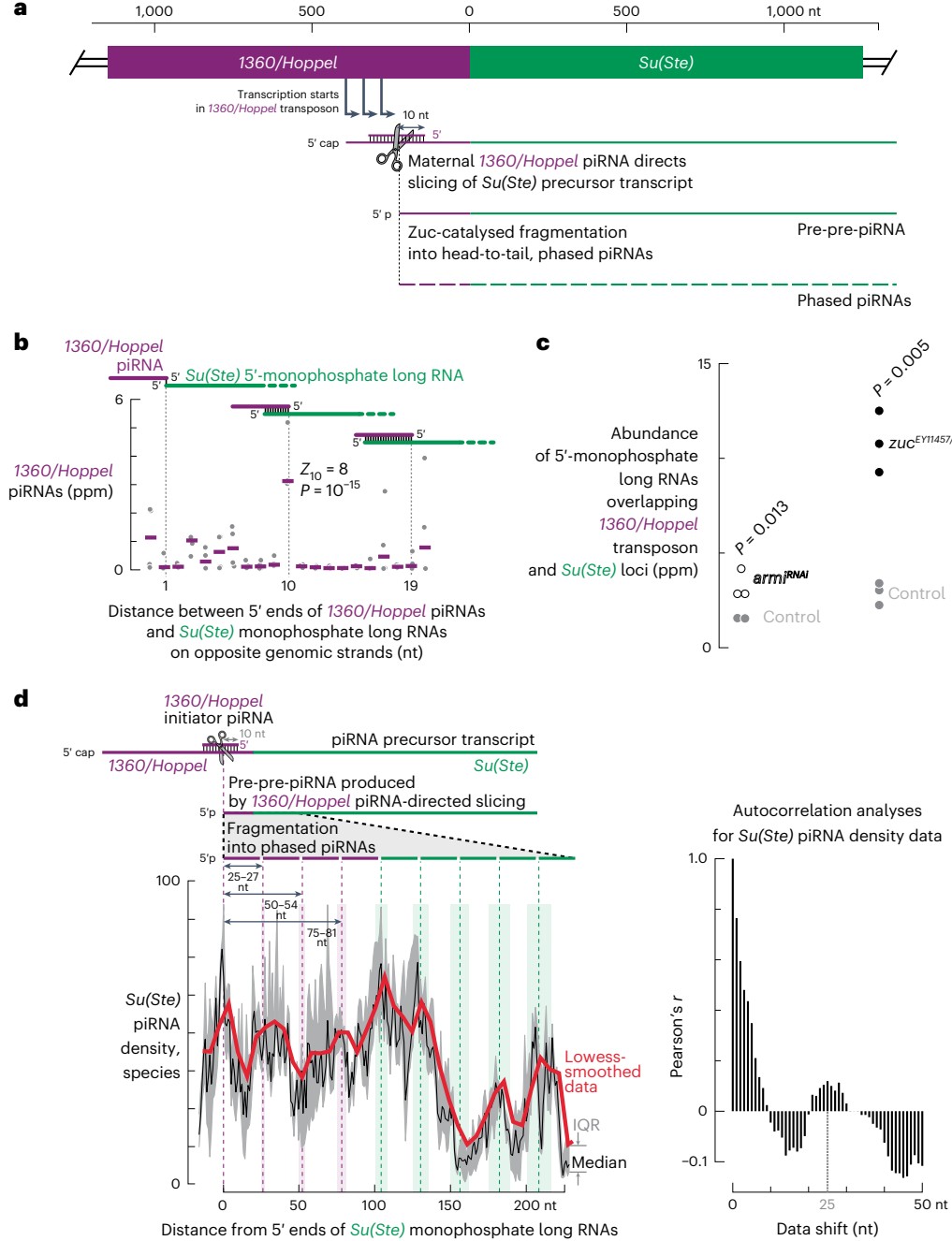

**Fig. 5 | Trigger piRNAs for phased *Su(Ste)* piRNA biogenesis in males.**
**a**, Model for initiation of phased biogenesis of *Su(Ste)* piRNAs by maternal *1360/Hoppel* piRNAs. **b**, Frequency of 0–20-nt overlaps between *Su(Ste)* 5′-monophosphorylated long RNAs and *1360/Hoppel* piRNAs on opposite genomic strands in control ($y^1w^{1118}/Y; nos$-gal4:VP16/TM2) testis. The standard score (number of standard deviations from the mean) and the corresponding *P* value (two-sided *Z*-test) of the 10-nt overlap ($Z_{10}$) is shown. Data are for all possible permutations of two small RNA datasets and two 5′-monophosphorylated long RNA datasets ($n = 2 × 2 = 4$). **c**, Change in steady-state abundance of 5′-monophosphorylated long RNA datasets in $nos{>}armi^{iRNAi}$ males ($n = 2$ for control; $n = 2$ for $nos{>}armi^{iRNAi}$) and in $zuc^{EYI1457/−}$ mutants ($n = 3$ for control; $n = 3$ for $zuc^{EYI1457/−}$); *P* values are shown for two-sided Mann–Whitney test. **d**, Left: metaplot of piRNA 5′-end density along *Su(Ste)*

long monophosphorylated RNAs in $nos{>}dpp$ testis. Data are for all possible permutations of small RNA and 5′-monophosphorylated long RNA datasets (12 permutations; $n = 3$ for 5′-monophosphorylated long RNA datasets; $n = 2$ for small RNA datasets used to identify putative cleavage products among 5′-monophosphorylated long RNAs; $n = 2$ for small RNA datasets used to plot piRNA density). Black line indicates the median; grey area shows IQR. Right: autocorrelation analyses of the median piRNA density data in the metaplot. In **b**, **c** and **d**, only 5′-monophosphorylated long RNAs that span a *Su(Ste)* locus and whose 5′ ends lie in the 100 nt flanking the upstream *1360/Hoppel* insertion were used for analyses (Supplementary Table 3; control testis: trial 1, 57 long RNAs; trial 2, 21 long RNAs; $nos{>}dpp$ testis: trial 1, 33 long RNAs; trial 2, 49 long RNAs; trial 3, 42 long RNAs). Source numerical data are available in source data.

with the 5′ ends of monophosphorylated RNAs from the genic loci *nos*, *bam* and *bgcn*, consistent with these RNAs being mRNA turnover intermediates (Extended Data Fig. 4a).

Among the *Su(Ste)*-derived, long, 5′-monophosphorylated RNAs overlapping the upstream *1360/Hoppel* transposon insertion, their 5′ ends most often corresponded to the scissile phosphate predicted

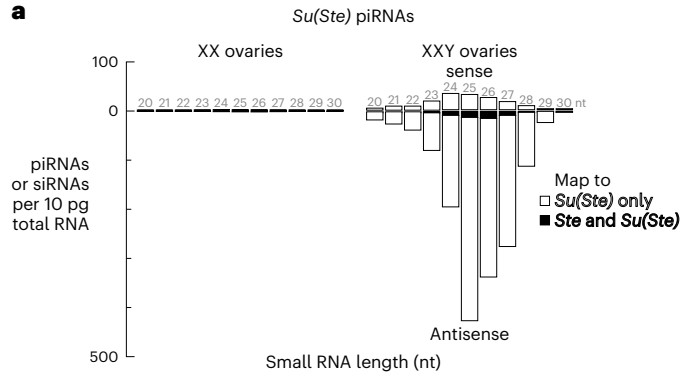

**Fig. 6 | Maternally deposited *Su(Ste)* piRNAs can rescue *Ste* repression in *armi*^RNAi^ male germline. a**, Length profile of *Su(Ste)*-derived (Supplementary Table 3) small RNAs in XX and XXY ovaries. The data are the mean from three independent biological samples. **b–e**, Representative images of smRNA-FISH for *Ste* (red) and antisense *Su(Ste)* (green) in the testes of control (**b**(i) *y¹w¹¹¹⁸/Y; nos-gal4:VP16/TM2*) and *nos>armi*^RNAi^ (**c**(i)) sons from XX mothers, and in testes of control (**d**(i)) and *nos>armi*^RNAi^ (**e**(i)) sons from XXY mothers. The asterisks indicate the hub. Red, *Ste* RNA; green, antisense *Su(Ste)* piRNA precursor; blue, DAPI. Magnified view of *Su(Ste)* piRNA precursor in situ hybridization signal at the apical tip of the testis is shown in **b**(ii), **c**(ii), **d**(ii) and **e**(ii). Arrowheads point to nuclear *Su(Ste)* transcripts; arrow points to cytoplasmic *Su(Ste)* RNA. Scale

bars, 20 µm (**b**(i)–**e**(i)) and 5 µm (**b**(ii)–**e**(ii)). Experiments were repeated three times with similar results. Source numerical data are available in source data. **f**, Top: model of developmental regulation of *Su(Ste)* piRNA biogenesis and *Ste* repression in males. *Su(Ste)* piRNA precursors are transcribed in early germ cells (GSCs and SGs), where they are processed to produce antisense *Su(Ste)* piRNAs by Armi- and Zuc-dependent, phased fragmentation. These *Su(Ste)* piRNAs are loaded into Aub and Ago3, which are later used in spermatocytes to cleave *Ste* transcripts. Phased fragmentation of *Su(Ste)* piRNA precursor is initiated by *1360/Hoppel* piRNAs deposited by XX mothers. Bottom: *Su(Ste)* piRNAs deposited by XXY mother can replace the need for Armi- and Zuc-dependent phased piRNA production.

from a complementary antisense *1360/Hoppel* piRNA ($Z_{10} = 8$, $P = 10^{-15}$; Fig. 5b). Our data therefore support the hypothesis that the majority of these monophosphorylated RNAs are pre-pre-piRNAs whose 5′ ends are made by *1360/Hoppel* piRNA-directed cleavage. Consistent with the idea that long RNAs from *42AB*, *petrel* and *Su(Ste)* are pre-pre-piRNAs processed by the phased biogenesis pathway, their steady-state abundance increased 1.7–5.4-fold when phased biogenesis in males was blocked in $zuc^{EYI1457/-}$ mutants or using *nos*-driven *armi^{RNAi}* (Fig. 5c and Extended Data Fig. 4b). By contrast, the abundance of 5′-monophosphorylated RNAs from *nos*, *bam* and *bgcn* did not change in $zuc^{EYI1457/-}$ or *nos>armi^{RNAi}* males (Extended Data Fig. 4b).

To examine *Su(Ste)* piRNA biogenesis in early male germ cells in more detail, we used *nos>dpp* males, in which SG overproliferate[67–69]. Among the ≥200-nt long, 5′-monophosphorylated RNAs from *nos>dpp* testis, we identified putative *Su(Ste)* pre-pre-piRNAs spanning both the *1360/Hoppel* and *Ste*-derived sequences that could have been produced by *1360/Hoppel* piRNA-guided slicing (Fig. 5d). The 5′ ends of *Su(Ste)* piRNAs concentrated in periodic peaks starting from *Su(Ste)* pre-pre-piRNA 5′ termini (Fig. 5d). Consistent with Zuc-catalysed fragmentation of pre-pre-piRNAs into tail-to-head pre-piRNAs, auto-correlation analyses showed that most piRNA 5′ ends lay at regular intervals, ~25–26 nt apart from each other (Fig. 5d). For *Su(Ste)*-derived pre-pre-piRNAs whose 5′ ends were in the last 100 nt of the *1360/Hoppel* sequence, most *Su(Ste)* piRNA 5′ ends occurred at ~25–27-nt intervals extending as far as ≥100 nt into the region of the *Su(Ste)* transcript that is antisense to *Ste* (Fig. 5d). Together, these data suggest that *1360/Hoppel* piRNAs slice *Su(Ste)* precursors to initiate 5′-to-3′ phased production of *Su(Ste)* piRNAs capable of silencing *Ste* mRNA.

### Su(Ste) piRNAs made in XXY females silence Ste in progeny

The remarkable stability of Argonaute-protected small RNAs[70,71] probably underlies the intergenerational inheritance of transposon-targeting piRNAs in animals with maternally deposited germ plasm. Similarly, our model assumes that piRNA•PIWI complexes deposited by mothers can cleave complementary RNAs in the germline of their sons. To experimentally test this assumption, we used XXY female flies to artificially deposit *Su(Ste)* piRNAs in oocytes. Y chromosome-encoded *Su(Ste)* piRNA precursors and *Su(Ste)* piRNAs were detected in XXY (2,700 ± 80 piRNAs per 10 pg total RNA) but not XX ovaries (30 ± 30 piRNAs per 10 pg total RNA; Fig. 6a, Extended Data Fig. 5 and Supplementary Table 2). These maternally produced *Su(Ste)* piRNAs were able to repress a *gfp-Ste* transgene in XXY females (Extended Data Fig. 6).

Strikingly, when *Su(Ste)* piRNA biogenesis was blocked in sons, maternal *Su(Ste)* piRNAs from XXY mother were sufficient to silence *Ste* in the testis: unlike *nos>armi^{RNAi}* males from XX mothers, *nos>armi^{RNAi}* sons derived from XXY females effectively repressed *Ste* (Fig. 6b–e and Extended Data Figs. 7 and 8). We conclude that maternal deposition of *Su(Ste)* piRNAs by XXY mothers suffices to silence *Ste* mRNA and bypasses the requirement for phased piRNA production pathway in early male germ cells.

## Discussion

The piRNA pathway is required for production of functional germ cells in animals. In species like *Drosophila*, whose germline is specified by maternally inherited determinants, the oocyte germ plasm contains piRNA•PIWI complexes that instruct their progeny to silence transposons antisense to the inherited piRNAs. Intergenerational continuity of the piRNA pathway in these species therefore relies on the continued passage of information through the germline. Such maternal inheritance is not possible for Y chromosome-encoded piRNAs, as females lack a Y chromosome. How can mothers instruct their sons to make piRNAs from precursors on the Y chromosome? Our data suggest that the *D. melanogaster* male germline relies on maternally deposited, transposon-derived piRNAs to trigger production of such *Su(Ste)* piRNAs antisense to *Ste* (Fig. 6f). The production of such

*Ste*-silencing piRNAs is possible because piRNA-directed cleavage of an RNA triggers the production of tail-to-head strings of piRNA via the phased piRNA biogenesis pathway. This model explains how fly males make piRNAs for which no homologous piRNA guides can be deposited by mothers. Our study also reveals that abundant *Su(Ste)* piRNAs are produced before the onset of transcription of their target, *Ste*. Such spatiotemporal separation may be required for effective repression of *Ste* mRNA.

In the fly germline, the proteins Rhino and Kipferl bind heterochromatic piRNA-producing loci and initiate transcription of precursor transcripts from both genomic strands[57,72–74]. Promoter-independent, RNA polymerase II transcription of these dual-strand piRNA clusters occurs throughout each locus, ignoring splice sites and polyadenylation sequences[75–78]. This atypical transcription strategy maximizes production of transposon-targeting piRNAs. *Su(Ste)* piRNA biogenesis in the male germline is unlikely to involve such non-canonical transcription of *Su(Ste)*. First, our smFISH experiments detected *Su(Ste)* transcripts from only one genomic strand. Second, loss of *rhi* in fly males has no effect on *Ste* silencing[56].

Taken together, our data suggest that the fly male germline has evolved a strategy that uses maternally supplied, transposon-derived piRNAs to generate Y chromosome-derived, *Su(Ste)* piRNAs that silence the selfish genetic element *Ste*. This strategy allows fly females to instruct their sons to produce piRNAs from sequences absent from the maternal genome. We speculate that this same mechanism may be used by mothers to protect their sons from selfish DNA in other animal species that deposit germline determinants in oocytes.

## Online content

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

## Methods

### Statistics and reproducibility

No statistical method was used to determine the sample size. For all biological samples, the maximum possible sample size ($n$ = 3–90) was chosen for each type of data ensuring that variability arising from all accountable sources was incorporated in the analyses (day of data collection, reagent lots, and experimenter). No data were excluded from the analyses. The experiments were not randomized, because this study did not involve treatment or exposure of animals to any agent. Instead, the goal of this work was to compare untreated wild-type/control flies and untreated mutant flies: all wild-type animals were compared with all mutant animals. The Investigators were not blinded to allocation during experiments and outcome assessment. Blinding was not performed during data collection, because methods used for data acquisition (smFISH, western blotting, qRT–PCR and high-throughput sequencing) are not influenced by the experimenter's knowledge of the fly genotype. Blinding was not performed during data analyses, because analyses were performed with the same automated algorithms and programming code. During analyses, wild-type control and mutant datasets are also easily identified and are directly compared with another.

### Fly husbandry and strains used

Flies (*D. melanogaster* strain w[1118]; 0–7 days old) were raised in standard Bloomington medium at 25 °C. The following stocks were obtained from the Bloomington Stock Center: *C(1)RM/C(X:Y)y[1]f[1]w[1]*, *armi[1]*, *armi[72.1]*, *aub[HN2]*, *aub[QC42]*, *zuc[EY11457]*, *Df(2L)BSC323*, *nos-gal4:VP16*, *bam-gal4:VP16*, *UAS-flag[3]-myc[6]-ago3* (ref. [80]), *UAS-gfp-aub*, *UAS-armi-gfp*, *UAS-dpp*, and RNAi lines for *armi*: TRIP.GL00254, *aub*: TRIP.GL00076, *ago3*: TRIP.HMC02938, *vasa*: TRIP.HMS00373, *zuc*: TRIP.GL00111. To generate *UAS-gfp-Ste* (*SteXh*:CG42398), cDNAs was synthetized (Invitrogen, sequence is provided in Supplementary Table 4), and inserted into *UAST-gfp* vector, after the *gfp* cDNA cassette, between BglII and XbaI sites. Transgenic lines carrying these transgenes were generated at BestGene.

To assay male fertility, a single male of indicated genotype (0–1 days old) was crossed to three *y[1]w[1118]* virgin females (0–2 days old) at room temperature. Flies were removed after 7 days, and the number of progeny was scored.

### Western blots

Testes (20 pairs per sample) were dissected and rinsed twice with 0.1 M phosphate buffer saline pH 7.2 (PBS), snap frozen and kept at −80 °C until use. Testes were homogenized in 100 µl (PBS), supplied with c0mplete protease inhibitor + ethylenediaminetetraacetic acid (Roche), and mixed with 100 µl of 2× Laemmli Sample Buffer (Bio-Rad). Cleared lysates were separated on a 12% Tris-glycine gel (Thermo Scientific) and transferred onto polyvinylidene fluoride membrane (Immobilon-P, Millipore). Mouse anti-α-Tubulin (clone 4.3; 1:3,000) (Walsh 1984) was obtained from the Developmental Studies Hybridoma Bank. The generation of polyclonal anti-Ste antibody (used at 1:10,000) was outsourced to Covance and was produced by immunizing guinea pigs with KLH-conjugated Ac-KPVIDSSSGLLYGDEKKWC (53–70 amino acids of Ste). Horseradish peroxidase-conjugated goat anti-mouse IgG (115-035-003; 1:10,000; Jackson ImmunoResearch Laboratories) and anti-guinea pig IgG (106-035-003; 1:10,000; Jackson ImmunoResearch Laboratories) secondary antibodies were used. The signals were detected by Pierce ECL Western Blotting Substrate enhanced chemiluminescence system (Thermo Scientific).

### smRNA-FISH

smRNA-FISH was conducted as described[61]. Testes from 2–3-day-old flies were dissected in 1× PBS, fixed in 4% formaldehyde in 1× PBS for 30 min, washed in PBS and permeabilized in 70% ethanol overnight at 4 °C. The following day, testes were rinsed with wash buffer (2× saline-sodium citrate and 10% formamide) and hybridized

overnight at 37 °C in hybridization buffer (2× saline-sodium citrate, 10% dextran sulfate (Sigma, D8906), 1 mg ml⁻¹ *Escherichia coli* tRNA (Sigma, R8759), 2 mM vanadyl ribonucleoside complex (NEB, S142), 0.5% bovine serum albumin (Ambion, AM2618) and 10% formamide). Following hybridization, samples were washed three times in wash buffer for 20 min each at 37 °C and mounted in VECTASHIELD with 4′,6-diamidino-2-phenylindole (DAPI, Vector Labs). Fluorescently labelled probes were added to the hybridization buffer to a final concentration of 100 nM. DNA oligo probes to detect *Ste* and *Su(Ste)* RNA were conjugated with Quasar 570, Cy3 or Cy5 fluorophores (Biosearch Technologies and IDT; for probe information, see Supplementary Table 5). Images were acquired using an upright Leica TCS SP8 confocal microscope with a 63× oil immersion objective lens (numerical aperture 1.4) and processed using ImageJ.

### qRT–PCR

Total RNA was isolated by Direct-zol RNA miniprep kit (Zymo Research) from biological triplicates of XY (100 testes per sample), XX or XXY gonads (60 ovaries per sample). Complementary DNA was generated by SuperScript III Reverse Transcriptase (Invitrogen) with random hexamer primers. qPCR of technical triplicates was performed using Power SYBR Green reagent (Applied Biosystems) and the following primer pairs. *Gapdh*: TAA ATT CGA CTC GAC TCA CGG T and CTC CAC CAC ATA CTC GGC TC, *act5C*: AAG TTG CTG CTC TGG TTG TCG and GCC ACA CGC AGC TCA TTG AG, *Su(Ste)*: TTC CGA AGT CAA GCG CTT CAA TG and GGA ATC TGT TTA ATT GCA ACA AC. $C_t$ values were normalized to *Gapdh* by the $2^{-\Delta\Delta C_t}$ method[81]. When calculating $\Delta C_t$ and $\Delta\Delta C_t$, standard deviations ($\sigma$) were propagated in Microsoft Excel 2013 using the formula $\sigma_x = \sqrt{\sigma_y^2 + \sigma_z^2}$.

### TaqMan small RNA analysis

The abundance of the following piRNAs were quantified by TaqMan small RNA custom assays (Thermo Fisher Scientific): *Su(Ste)-4* piRNA (target sequence: UCU CAU CGU CGU AGA ACA AGC CCG A), the most abundant *Su(Ste)* piRNA[54]; *piR-dme-1643 piRNA* (piRBase nomenclature), target sequence: (TAA AGC GTT GTT TTG TGC TAT ACC C), a piRNA we found to be highly abundant in the ovary based on analysis of earlier small RNA sequencing data, and 2S ribosomal RNA (rRNA) (target sequence: UGC UUG GAC UAC AUA UGG UUG AGG GUU GUA), which we utilized in this study as control. Total RNA was isolated from biological triplicates of XX and XXY ovaries (60 per sample) by Direct-zol miniprep kit (Zymo Research). Reverse transcription and qPCR were performed following the manufacturer's protocol using TaqMan MicroRNA Reverse Transcription Kit, and TaqMan Universal PCR Master Mix II, No UNG (Thermo Fisher Scientific). qPCRs were performed in technical triplicates with the appropriate controls. $C_t$ values were normalized to 2S rRNA levels by the $2^{-\Delta\Delta C_t}$ method[81]. When calculating $\Delta C_t$ and $\Delta\Delta C_t$, standard deviations ($\sigma$) were propagated in Microsoft Excel 2013 using the formula $\sigma_x = \sqrt{\sigma_y^2 + \sigma_z^2}$.

### Small RNA-seq library preparation and analyses

Total RNA from fly ovaries or testis was extracted using the mirVana miRNA isolation kit (Thermo Fisher, AM1560). Small RNA libraries were constructed as described[82] with modifications. Briefly, before library preparation, a spike-in RNA mix, an equimolar mix of six synthetic 5′-phosphorylated RNA oligonucleotides (/phos/UGC UAG UCU UAU CGA CCU CCU CAU AG, /phos/UGC UAG UCU UCG AUA CCU CCU CAU AG, /phos/UGC UAG UCU UGU CAC GAA CCU CAU AG, /phos/UGC UAG UUA UCG ACC UUC AUA G, /phos/UGC UAG UUC GAU ACC UUC AUA G, /phos/UGC UAG UUG UCA CGA AUC AUA G), was added to each RNA sample to enable absolute quantification of small RNAs (Supplementary Table 6). To reduce ligation bias and eliminate PCR duplicates, the 3′ and 5′ adaptors both contained nine random nucleotides at their 5′ or 3′ ends, respectively (see below) and 3′ adaptor ligation reactions contained 25% (w/v) PEG-8000 final concentration (f.c.).

Total RNA was run through a 15% denaturing urea–polyacrylamide gel (National Diagnostics) to isolate 15–29-nt small RNAs and remove the 30-nt 2S rRNA. After overnight elution in 0.4 M NaCl followed by ethanol precipitation, small RNAs were oxidized (to clone only 2′-*O*-methylated siRNAs and piRNAs) in 40 μl 200 mM sodium periodate, 30 mM borax, 30 mM boric acid (pH 8.6) at 25 °C for 30 min. After ethanol precipitation, small RNAs were ligated to 25 pmol 3′ DNA adapter with adenylated 5′ and dideoxycytosine-blocked 3′ ends (/rApp/NNN GTC NNN TAG NNN TGG AAT TCT CGG GTG CCA AGG/ddC/) in 30 μl 50 mM Tris–HCl (pH 7.5), 10 mM MgCl$_2$, 10 mM dithiothreitol (DTT) and 25% (w/v) PEG-8000 (NEB) with 600 U homemade T4 Rnl2tr K227Q at 16 °C overnight. After ethanol precipitation, the 50–90-nt (14–54-nt small RNA + 36-nt 3′ unique molecular identifier adapter) 3′-ligated product was purified from a 15% denaturing urea–polyacrylamide gel (National Diagnostics). After overnight elution in 0.4 M NaCl followed by ethanol precipitation, the 3′-ligated product was denatured in 13 μl water at 90 °C for 60 s, 1 μl 10 μM anti-2S oligo (TAC AAC CCT CAA CCA TAT GTA GTC CAA GCA-/3′ C3 Spacer/; to suppress the ligation of 2S rRNA) and 1 μl 50 μM RT primer (CCT TGG CAC CCG AGA ATT CCA; to suppress the formation of 5′-adapter:3′-adapter dimers) were added and annealed at 65 °C for 5 min. The resulting mix was then ligated to a mixed pool of equimolar amount of two 5′ RNA adapters (to increase nucleotide diversity at the 5′ end of the sequencing read: GUU CAG AGU UCU ACA GUC CGA CGA UCN NNC GAN NNU CAN NN and GUU CAG AGU UCU ACA GUC CGA CGA UCN NNA UCN NNA GUN NN) in 20 μl 50 mM Tris–HCl (pH 7.8), 10 mM MgCl$_2$, 10 mM DTT, 1 mM ATP with 20 U of T4 RNA ligase (Thermo Fisher, EL0021) at 25 °C for 2 h. The ligated product was precipitated with ethanol, and cDNA synthesis was performed in 20 μl at 42 °C for 1 h using AMV reverse transcriptase (NEB, M0277) and 5 μl RT reaction was amplified in 25 μl using AccuPrime Pfx DNA polymerase (Thermo Fisher, 12344024; 95 °C for 2 min, 15 cycles of: 95 °C for 15 s, 65 °C for 30 s, 68 °C for 15 s; forward primer: AAT GAT ACG GCG ACC ACC GAG ATC TAC ACG TTC AGA GTT CTA CAG TCC GA; reverse primer: CAA GCA GAA GAC GGC ATA CGA GAT XXX XXX GTG ACT GGA GTT CCT TGG CAC CCG AGA ATT CCA, where XXXXXX represents the 6-nt sequencing barcode). Finally, the PCR product was purified in a 2% agarose gel. Small RNA-seq libraries samples were sequenced using a NextSeq 550 (Illumina) to obtain 79 nt, single-end reads.

The 3′ adapter (TGG AAT TCT CGG GTG CCA AGG) was removed with fastx toolkit (v0.0.14), PCR duplicates were eliminated as described[83], and rRNA matching reads were removed with bowtie (parameter -v 1; v1.0.0) against *D. melanogaster* set in SILVA database[84]. Deduplicated and filtered data were analysed with Tailor[85] to account for non-templated tailing of small RNAs. Sequences of synthetic RNA spike-in oligonucleotides were identified allowing no mismatches with using bowtie (parameter -v 0; v1.0.0), and the absolute abundance of small RNAs calculated. The background for $Z_{10}$ calculation was all displayed data except position 10.

## RNA-seq library preparation and analyses
Total RNA from sorted germ cells was extracted using the mirVana miRNA isolation kit (ThermoFisher, AM1560). Before library preparation, to remove rRNA, 1 μg total RNA was hybridized in 10 μl to a pool of 186 rRNA antisense oligos (0.05 μm f.c. each) in 10 mM Tris–HCl (pH 7.4), 20 mM NaCl by heating to 95 °C, cooling at −0.1 °C s$^{-1}$ to 22 °C, and incubating at 22 °C for 5 min. RNase H (10 U; Lucigen, H39500) was added and the mixture incubated at 45 °C for 30 min in 20 μl containing 50 mM Tris–HCl (pH 7.4), 100 mM NaCl and 20 mM MgCl$_2$. The reaction volume was adjusted to 50 μl with 1× TURBO DNase buffer (ThermoFisher, AM2238) and then incubated with 4 U TURBO DNase (ThermoFisher, AM2238) for 20 min at 37 °C. Next, RNA was purified using RNA Clean & Concentrator-5 (Zymo Research, R1016) to retain ≥200-nt RNAs, followed by the stranded, dUTP-based RNA-seq

protocol described in ref. 86 using adapters with unique molecular identifiers from ref. 83. RNA-seq libraries were sequenced using a NextSeq 550 (Illumina) to obtain 79 + 79 nt, paired-end reads.

RNA-seq analysis was performed using piPipes for genomic alignment[87]. Briefly, before starting piPipes, sequences were reformatted to extract unique molecular identifiers[83]. The reformatted reads were then aligned to rRNA using bowtie2 (v2.2.0). Unaligned reads were mapped to the dm6 assembly using STAR (v2.3.1), and PCR duplicates removed[83]. Transcript abundance was calculated using StringTie (v1.3.4). Differential expression analysis was performed using DESeq2 (v1.18.1).

## Cloning and sequencing of 5′-monophosphorylated long RNAs
Total RNA from fly ovaries or testis was extracted using mirVana miRNA isolation kit (ThermoFisher, AM1560) and used to prepare a library of 5′-monophosphorylated long RNAs as described[82] with modifications. Briefly, to deplete rRNA, 1 μg total RNA was hybridized in 10 μl to a pool of rRNA antisense oligos (0.05 μm f.c. each) in 10 mM Tris–HCl (pH 7.4), 20 mM NaCl by heating the mixture to 95 °C, cooling it at −0.1 °C s$^{-1}$ to 22 °C, and incubating at 22 °C for 5 min. RNase H (10 U; Lucigen, H39500) was added and the mixture incubated at 45 °C for 30 min in 20 μl containing 50 mM Tris–HCl (pH 7.4), 100 mM NaCl and 20 mM MgCl$_2$. The reaction volume was adjusted to 50 μl with 1× TURBO DNase buffer (ThermoFisher, AM2238) and then incubated with 4 U TURBO DNase (ThermoFisher, AM2238) for 20 min at 37 °C. Next, RNA was purified using RNA Clean & Concentrator-5 (Zymo Research, R1016) to retain ≥200-nt fragments. RNA was then ligated to a mixed pool of equimolar amounts of two 5′ RNA adapters (to increase nucleotide diversity at the 5′ end of the sequencing read: GUU CAG AGU UCU ACA GUC CGA CGA UCN NNC GAN NNU CAN NN and GUU CAG AGU UCU ACA GUC CGA CGA UCN NNA UCN NNA GUN NN) in 20 μl of 50 mM Tris–HCl (pH 7.8), 10 mM MgCl$_2$, 10 mM DTT and 1 mM ATP with 60 U of High Concentration T4 RNA ligase (NEB, M0437M) at 16 °C overnight. The ligated product was isolated using RNA Clean & Concentrator-5 (Zymo Research, R1016) to retain ≥200-nt RNAs and reverse transcribed in 25 μl with 50 pmol RT primer (GCA CCC GAG AAT TCC ANN NNN NNN) using SuperScript III (ThermoFisher, 18080093). After purification with 50 μl Ampure XP beads (Beckman Coulter, A63880), cDNA was PCR amplified using NEBNext High-Fidelity (NEB, M0541; 98 °C for 30 s; four cycles of: 98 °C for 10 s, 59 °C for 30 s, 72 °C for 12 s; six cycles of: 98 °C for 10 s, 68 °C for 10 s, 72 °C for 12 s; 72 °C for 3 min; with the following primers: CTA CAC GTT CAG AGT TCT ACA GTC CGA and GCC TTG GCA CCC GAG AAT TCC A). PCR products between 200 bp and 400 bp were isolated with a 1% agarose gel, purified with QIAquick Gel Extraction Kit (Qiagen, 28706), and amplified again with NEBNext High-Fidelity (NEB, M0541; 98 °C for 30 s; 3 cycles of: 98 °C for 10 s, 68 °C for 30 s, 72 °C for 14 s; six cycles of: 98 °C for 10 s, 72 °C for 14 s; 72 °C for 3 min; forward primer: AAT GAT ACG GCG ACC ACC GAG ATC TAC ACG TTC AGA GTT CTA CAG TCC GA; reverse primer: CAA GCA GAA GAC GGC ATA CGA GAT XXX XXX GTG ACT GGA GTT CCT TGG CAC CCG AGA ATT CCA, where XXXXXX represents the 6-nt sequencing barcode). The PCR product was purified in a 1% agarose gel and sequenced using a NextSeq 550 to obtain 79 + 79 nt, paired-end reads.

Sequencing data was aligned to the fly genome (dm6) with piPipes[87]. Briefly, before starting piPipes, sequences were reformatted to remove the degenerate portion of the 5′ adapter (nucleotides 1–15 of read 1). The reformatted reads were then aligned to fly rRNA using bowtie2 (v2.2.0). Unaligned reads were mapped to the fly genome (dm6) using STAR (v2.3.1), alignments with soft clipping of ends were removed with SAMtools (v1.0.0), and reads with the same 5′ end were merged to represent a single 5′-monophosphorylated RNA species.

## Reporting summary
Further information on research design is available in the Nature Portfolio Reporting Summary linked to this article.

## Data availability

Sequencing data generated in this study have been deposited in the National Center for Biotechnology Information Short Read Archive database under accession code PRJNA879723. Fly genome sequence and annotation (build dm6/BDGP6.22 release 98) used in this study were downloaded from Ensembl at ftp://ftp.ensembl.org/pub/release-98/fasta/drosophila_melanogaster/ and ftp://ftp.ensembl.org/pub/release-98/gtf/drosophila_melanogaster/; fly rRNA sequences were downloaded from SILVA rRNA database at https://www.arb-silva.de/. Source data are provided with this paper. All other data supporting the findings of this study are available from the corresponding authors upon request.

## Code availability

Code used in this work is deposited at https://github.com/ildargv/Venkei_et_al_2023.

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

## Acknowledgements

We thank the Bloomington *Drosophila* Stock Center and the Developmental Studies Hybridoma Bank for reagents. We thank Z. Zhang and N. Lau for their helpful discussions and the Yamashita lab members for comments on the paper. The research was supported by the Howard Hughes Medical Institute (Y.M.Y., P.D.Z. and S.E.J.), National Institute of Health (NIH R01 HD109667 to J.K.K., R35 GM136275 to P.D.Z., R01GM097363 to A.A.A. and R35 GM130272 to S.E.J.) and the Whitehead Institute for Biomedical Research (Y.M.Y.).

## Author contributions

Z.G.V. and Y.M.Y. conceived the project. Z.G.V., I.G., S.E.J., J.K.K., P.D.Z. and Y.M.Y. designed experiments and interpreted the results. Z.G.V., I.G., A.B., C.B., J.M.F. and Y.M.Y. conducted experiments. Z.G.V., I.G., M.R.S., C.P.C., T.W.W., G.W.B. and S.F. analysed data. P.C. and A.A.A. contributed critical information in the course of the investigation. Z.G.V., I.G., Y.M.Y. and P.D.Z. wrote and edited the paper with the input from other authors. Y.M.Y. and P.D.Z. supervised the research.

## Competing interests

The authors declare no competing interests.

## Additional information

**Extended data** is available for this paper at https://doi.org/10.1038/s41556-023-01227-4.

**Correspondence and requests for materials** should be addressed to Phillip D. Zamore or Yukiko M. Yamashita.

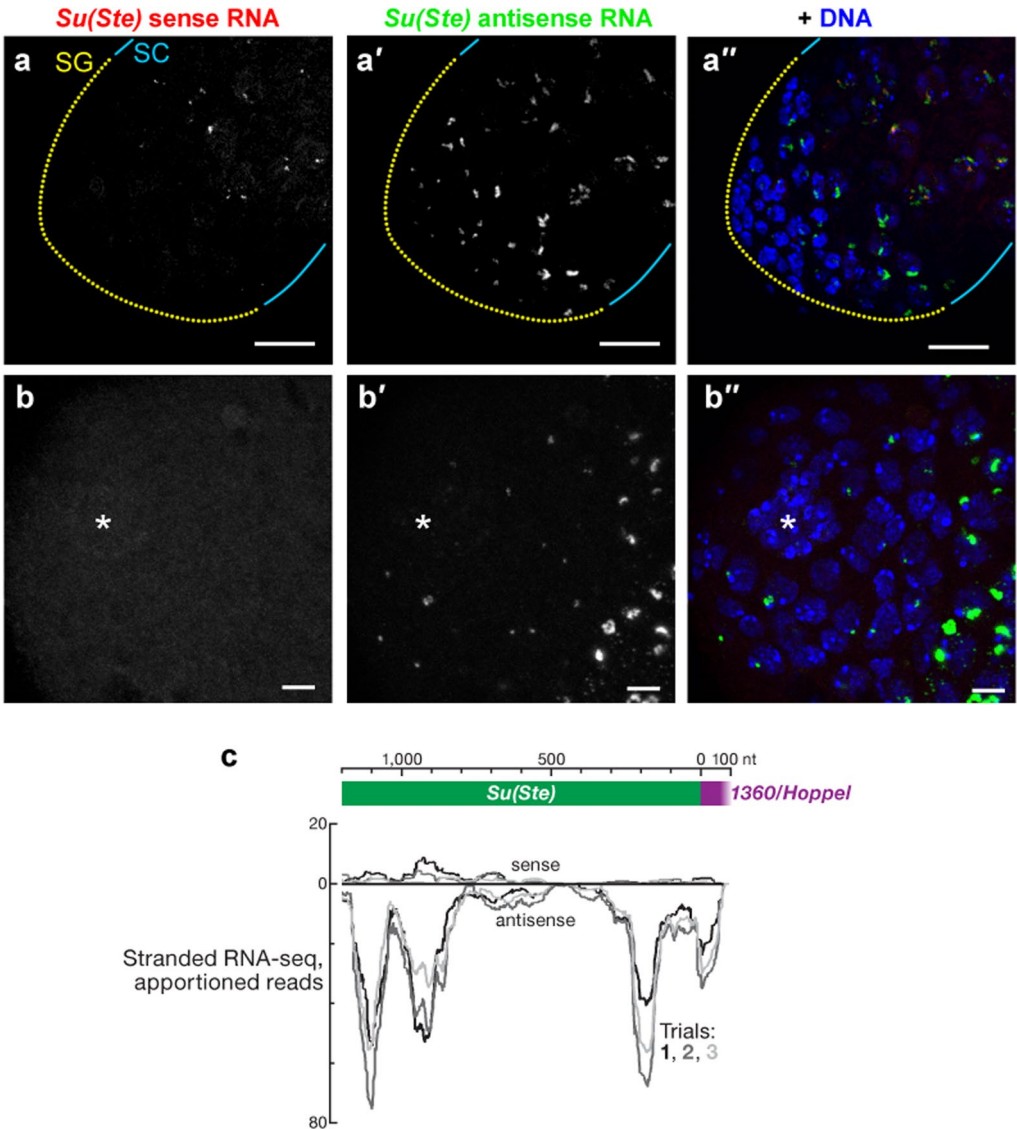

**Extended Data Fig. 1 | Expression of sense and anti-sense _Su(Ste)_ precursor RNA in testis. a, b**, smRNA-FISH for sense (red) and antisense (green) _Su(Ste)_ precursor transcripts in the apical tip of the testis (a, lower magnification; b, higher magnification, not the same tissue). Hub (*), DAPI (blue). Bar: 20 μm in a, 5 μm in b. **c**, Metaplot of stranded RNA-seq coverage in _Su(Ste)_ loci (Supplementary Table 1). The data are shown for three independent biological samples. Source numerical data are available in source data.

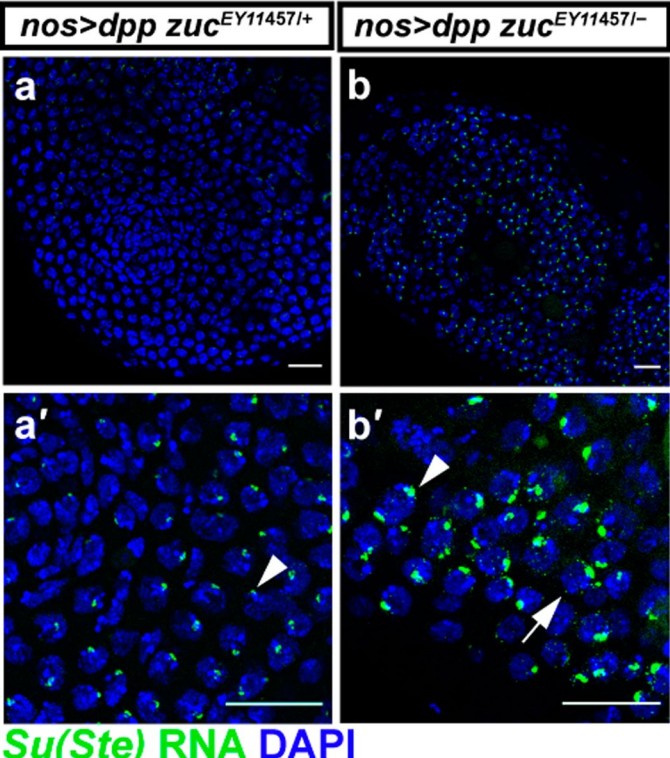

**Extended Data Fig. 2 | *Su(Ste)* piRNA precursor accumulates in *zuc* mutant testis. a, b**, Representative images of smRNA-FISH for antisense *Su(Ste)* precursor RNAs (green) in adult testes of the indicated genotypes. Arrows point to cytoplasmic precursor RNAs; arrowheads point to nuclear transcripts. DAPI (blue), bars 20 μm (a and b) and 5 μm (a′ and b′). Experiments were repeated three times with similar results.

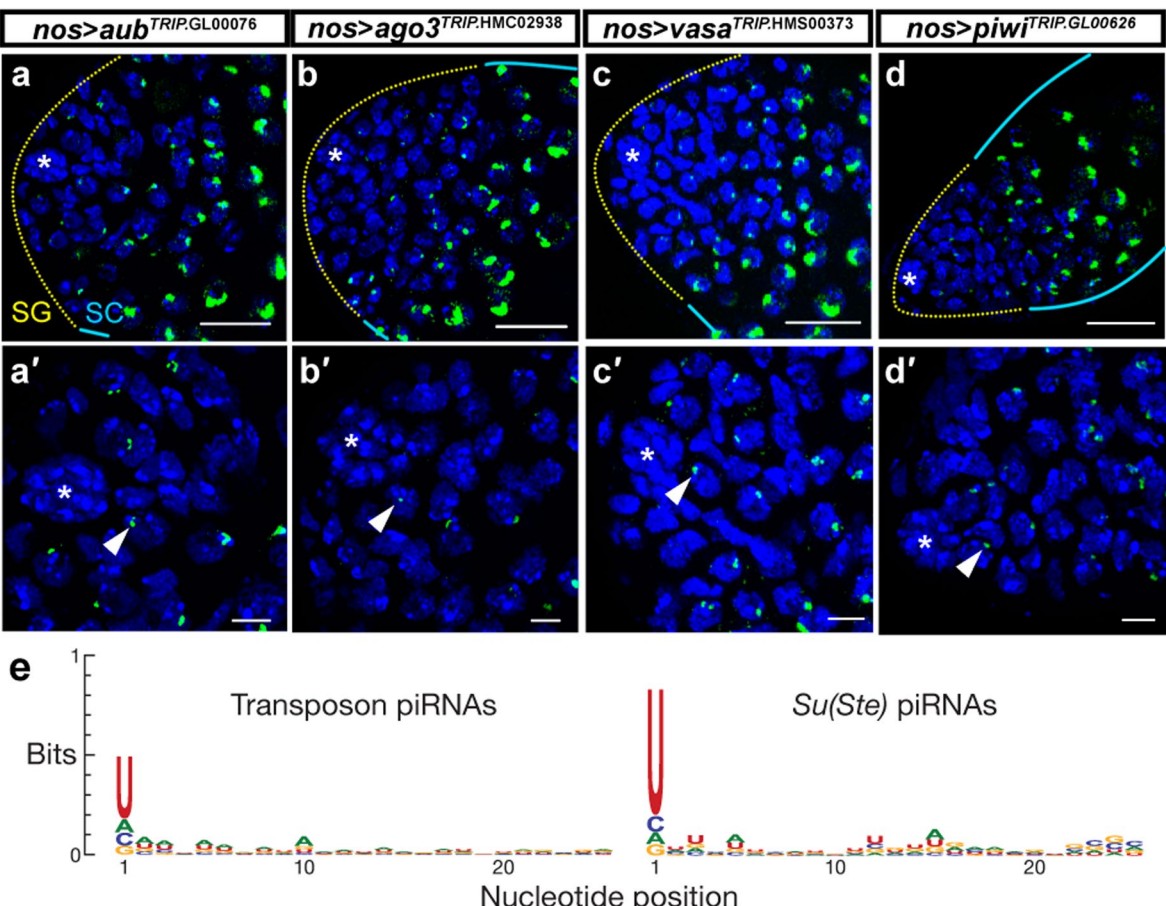

**Extended Data Fig. 3 | *Su(Ste)* piRNA precursor is not upregulated upon knockdown of *aub, vasa, ago3,* or *piwi*. a–d,** *Su(Ste)* piRNA precursor transcript testes of the indicated genotypes. Magnified regions of the niche are shown in a′, b′, c′, d′. The region of GSCs/SGs is indicated by a yellow dotted line, SC region by cyan lines. Arrowheads point to nuclear transcripts. Hub (*), DAPI (blue), bars 20 μm (a–d) and 5 μm (a′–d′). **e,** Bias in nucleotide composition (sequence logo) of transposon- and *Su(Ste)*-derived (Supplementary Table 1) piRNAs in control testis from 0–5-day-old $y^1w^{1118}$/Y; *nos-gal4:VP16* males. The data are the mean of two independent biological samples. Source numerical data are available in source data.

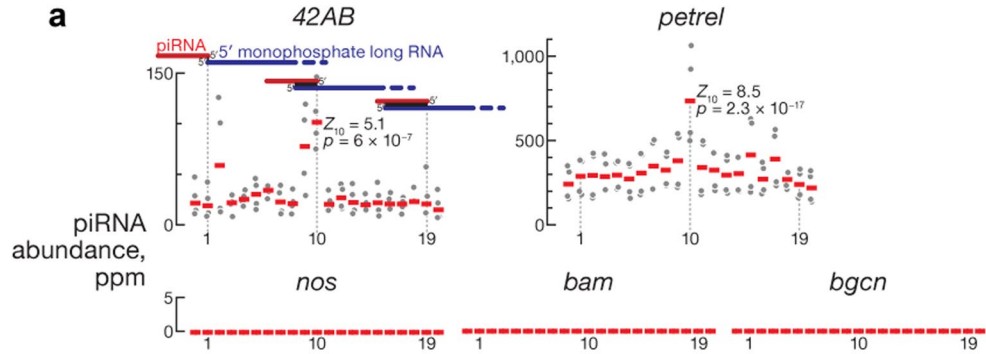

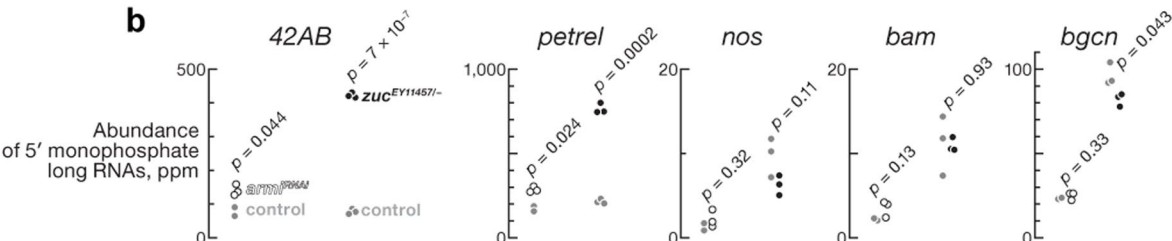

**Extended Data Fig. 4 | Long 5′ monophosphorylated RNAs from piRNA producing loci *42AB* and *petrel* and from *nos*, *bam*, *bgcn* genic loci. a**, Frequency of 0–20-nt overlaps between 5′ monophosphorylated long RNAs and piRNAs on opposite genomic strands in control testis from 0–5-day-old $y^1 w^{1118}/Y; nos$-$gal4$:$VP16$ males. The standard score (number of standard deviations from the mean) and the corresponding $p$ value (two-sided $Z$-test) of the 10-nt overlap ($Z_{10}$) is shown. Data are for all possible permutations of two small RNA data sets and two 5′ monophosphorylated long RNA data sets ($n = 2 \times 2 = 4$). **b**, Change in steady-state abundance of 5′ monophosphorylated long RNA data sets in *nos>armi^{RNAi}* males ($n = 2$ for control; $n = 2$ for *nos>armi^{RNAi}*) and in *zuc^{EY11457/−}* mutants ($n = 3$ for control; $n = 3$ for *zuc^{EY11457/−}*); p values are shown for the two-sided Mann-Whitney test. Source numerical data are available in source data.

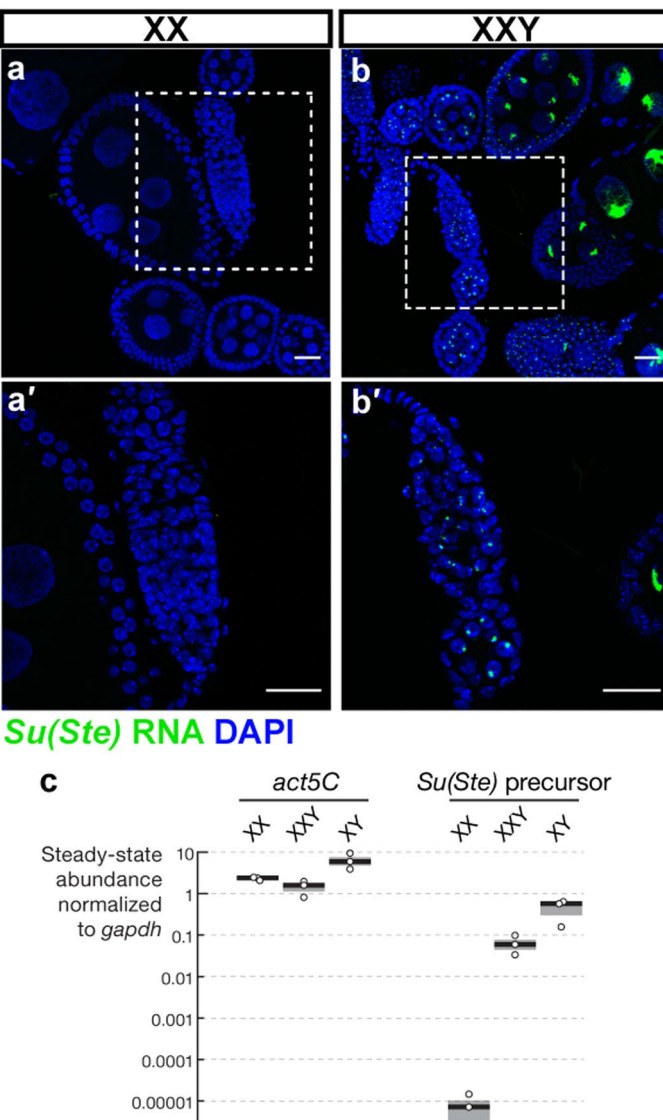

**Extended Data Fig. 5 | *Su(Ste)* precursor transcripts and piRNAs in XXY ovaries. a, b**, Germaria and early egg chambers of XX (a) and XXY (b) females with magnified inserts of germaria shown in a' and b'. Antisense *Su(Ste)* piRNA precursor transcript (green), DAPI (blue), bars 20 μm. **c**, Relative abundance of *act5C* mRNA and antisense *Su(Ste)* piRNA precursor transcript in XX and XXY ovaries, and in XY testis, determined by qRT-PCR, normalized to *Gapdh* ($n = 3$). Boxplots show the median and interquartile range (IQR). Source numerical data are available in source data.

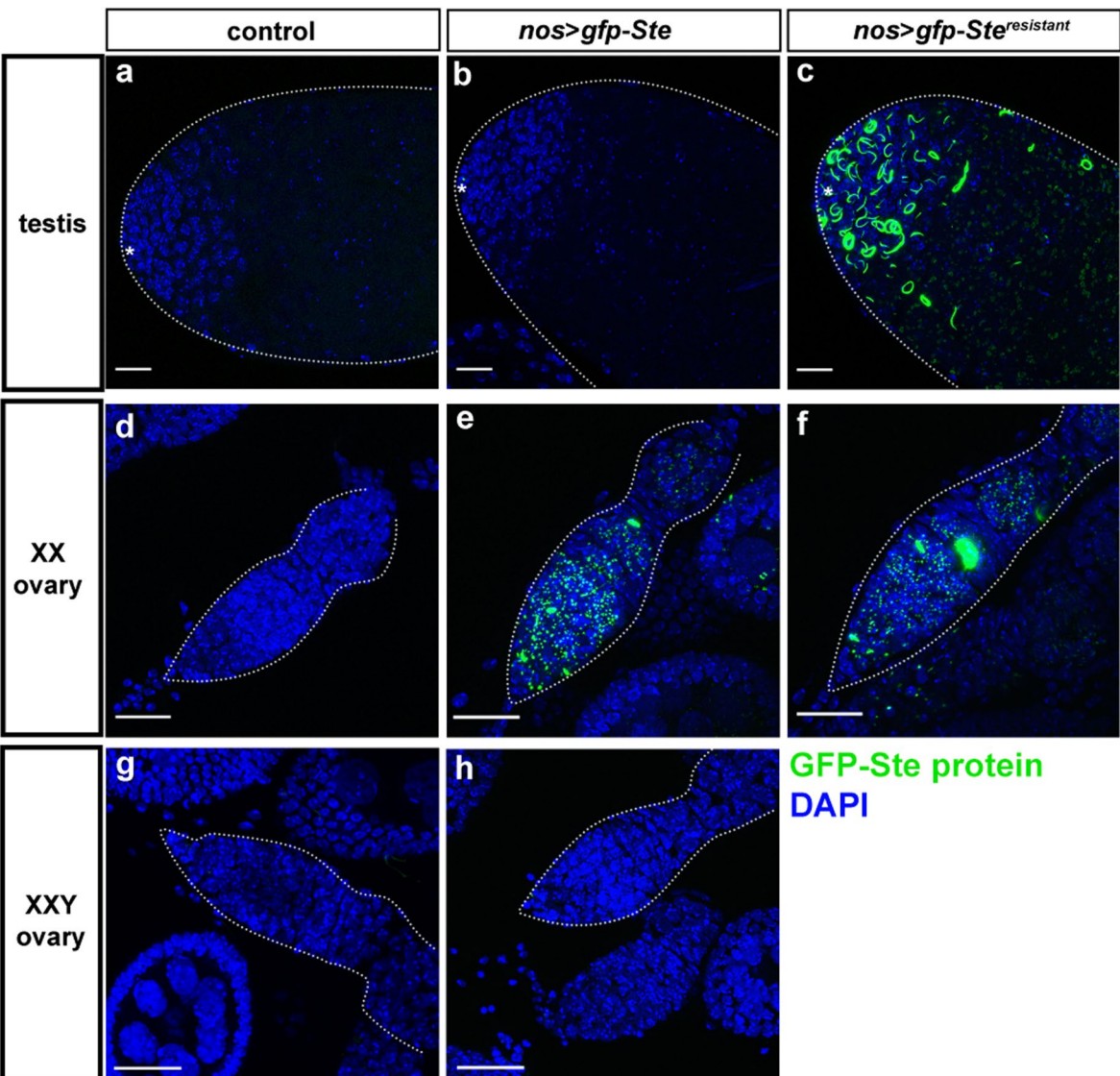

**Extended Data Fig. 6 | *gfp-Ste* reporter is silenced in the ovary of XXY females.** Representative images of GFP (green) in testis from XY males (a–c) or germaria from XX (d–f) or XXY (g–h) females. DAPI (blue), bars 20 μm. Asterisk indicates the hub in a–c. Experiments were repeated three times with similar results.

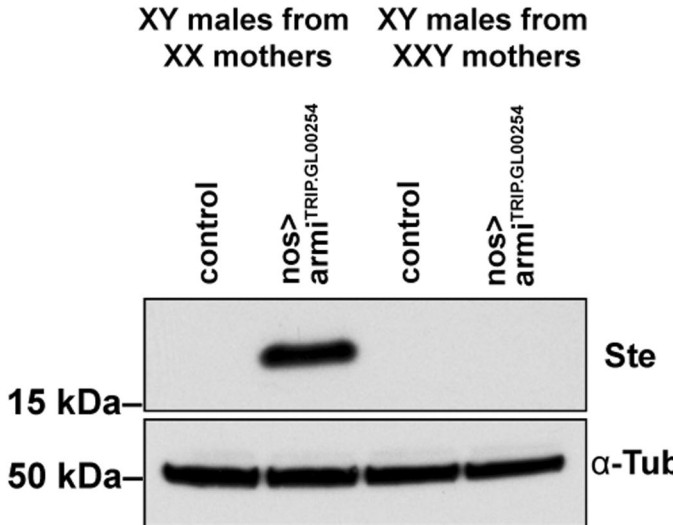

**Extended Data Fig. 7 | Repression of Ste protein in *armi*[RNAi] males from XXY mothers.** Representative images of Anti-Ste and anti-Tubulin Western blotting of testes from the indicated genotypes. Source numerical data and unprocessed blots are available in source data. Experiments were repeated twice with indistinguishable results.

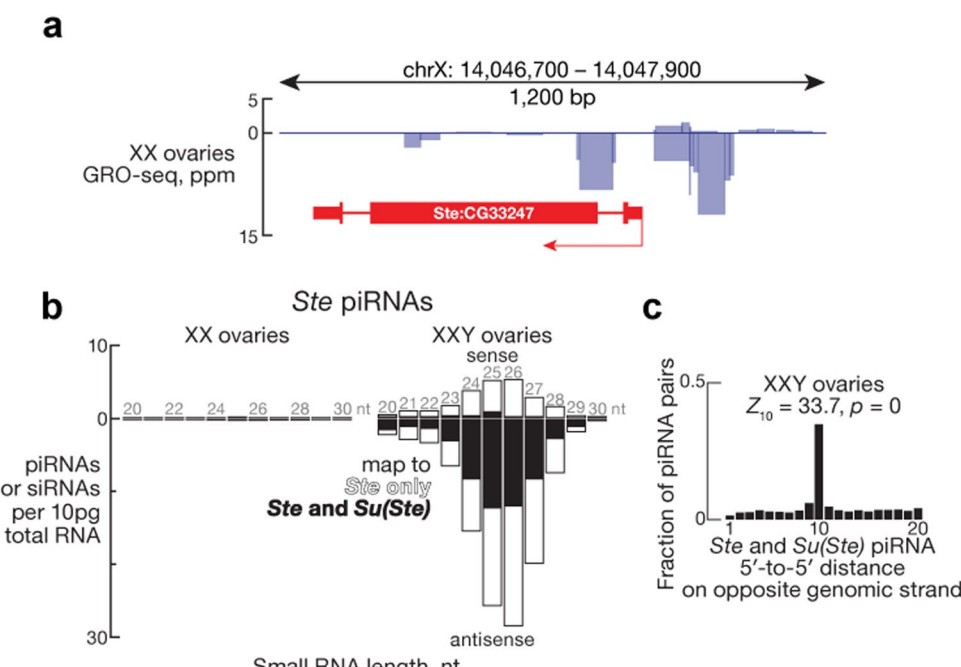

**Extended Data Fig. 8 | *Su(Ste)* piRNAs make *Ste* piRNAs in XXY ovaries.**
**a**, Nascent transcripts (GRO-seq) at a *Ste* locus in $w^1$ XX ovaries. Data are from ref. 79 for all (uniquely and multiply mapping) reads without apportioning to other *Ste* loci. **b**, Length profile of *Ste*-derived small RNAs in XXY ovaries. The data are the mean of three independent biological samples. **c**, Ping-pong signature— that is, frequent 10-nt overlap on opposite genomic strands—between *Su(Ste)* and *Ste*-derived piRNAs in XXY ovaries. The data are the mean of three independent biological samples. The standard score (number of standard deviations from the mean) and the corresponding $p$ value (two-sided $Z$-test) of the 10-nt overlap ($Z_{10}$) is shown. Source numerical data are available in source data.

# Reporting Summary

## Statistics

For all statistical analyses, confirm that the following items are present in the figure legend, table legend, main text, or Methods section.

| n/a | Confirmed | |
|---|---|---|
| ☐ | ☒ | The exact sample size (*n*) for each experimental group/condition, given as a discrete number and unit of measurement |
| ☐ | ☒ | A statement on whether measurements were taken from distinct samples or whether the same sample was measured repeatedly |
| ☐ | ☒ | The statistical test(s) used AND whether they are one- or two-sided<br>*Only common tests should be described solely by name; describe more complex techniques in the Methods section.* |
| ☒ | ☐ | A description of all covariates tested |
| ☒ | ☐ | A description of any assumptions or corrections, such as tests of normality and adjustment for multiple comparisons |
| ☐ | ☒ | A full description of the statistical parameters including central tendency (e.g. means) or other basic estimates (e.g. regression coefficient) AND variation (e.g. standard deviation) or associated estimates of uncertainty (e.g. confidence intervals) |
| ☐ | ☒ | For null hypothesis testing, the test statistic (e.g. *F*, *t*, *r*) with confidence intervals, effect sizes, degrees of freedom and *P* value noted<br>*Give P values as exact values whenever suitable.* |
| ☒ | ☐ | For Bayesian analysis, information on the choice of priors and Markov chain Monte Carlo settings |
| ☒ | ☐ | For hierarchical and complex designs, identification of the appropriate level for tests and full reporting of outcomes |
| ☐ | ☒ | Estimates of effect sizes (e.g. Cohen's *d*, Pearson's *r*), indicating how they were calculated |

*Our web collection on statistics for biologists contains articles on many of the points above.*

## Software and code

Policy information about availability of computer code

| Data collection | Illumina NextSeq 550, Leica TCS SP8 confocal microscope |
|---|---|
| Data analysis | fastx toolkit (v0.0.14); bowtie2 (v2.2.0); STAR (v2.3.1); StringTie (v1.3.4); bowtie (v1.0.0); SAMtools (v1.0.0); DESeq2 (v1.18.1); Microsoft Excel 2013; ImageJ |

For manuscripts utilizing custom algorithms or software that are central to the research but not yet described in published literature, software must be made available to editors and reviewers. We strongly encourage code deposition in a community repository (e.g. GitHub). See the Nature Portfolio guidelines for submitting code & software for further information.

## Data

Policy information about availability of data

All manuscripts must include a data availability statement. This statement should provide the following information, where applicable:
- Accession codes, unique identifiers, or web links for publicly available datasets
- A description of any restrictions on data availability
- For clinical datasets or third party data, please ensure that the statement adheres to our policy

Sequencing data generated in this study have been deposited in the National Center for Biotechnology Information Short Read Archive database under accession code PRJNA879723 and are available at https://www.ncbi.nlm.nih.gov/bioproject/PRJNA879723/. Fly genome sequence and annotation (build dm6/BDGP6.22 release 98) used in this study were downloaded from Ensembl at ftp://ftp.ensembl.org/pub/release-98/fasta/drosophila_melanogaster/ and ftp://ftp.ensembl.org/

## Human research participants

Policy information about studies involving human research participants and Sex and Gender in Research.

| | |
|---|---|
| Reporting on sex and gender | N/A |
| Population characteristics | N/A |
| Recruitment | N/A |
| Ethics oversight | N/A |

Note that full information on the approval of the study protocol must also be provided in the manuscript.

# Field-specific reporting

Please select the one below that is the best fit for your research. If you are not sure, read the appropriate sections before making your selection.

☒ Life sciences          ☐ Behavioural & social sciences          ☐ Ecological, evolutionary & environmental sciences

For a reference copy of the document with all sections, see nature.com/documents/nr-reporting-summary-flat.pdf

# Life sciences study design

All studies must disclose on these points even when the disclosure is negative.

| | |
|---|---|
| Sample size | No statistical method was used to determine the sample size. For all biological samples, the maximum possible sample size (n = 3–90) was chosen for each type of data ensuring that variability arising from all accountable sources was incorporated in the analyses (day of data collection, reagent lots, experimenter). |
| Data exclusions | No data were excluded from the analyses. |
| Replication | All data were collected during independent trials (n = 3) conducted on separate days. All attempts at replication were successful. When using several types of data for analyses, all possible permutations of samples were analyzed (e.g., 3 small RNA sequencing × 3 5' monophosphorylated RNA sequencing data sets produced 9 permutations). All attempts at replication were successful. |
| Randomization | This study did not involve treatment or exposure of animals to any agent. Instead, the goal of this work was to compare untreated wild-type/control flies and untreated mutant flies: all wild-type animals were compared to all mutant animals. Therefore, randomization is not relevant to this study. |
| Blinding | Blinding is not relevant to this study. Blinding was not performed during data collection, because methods used for data acquisition (smFISH, Western blotting, RT-qPCR, high-throughput sequencing) are not influenced by the experimenter's knowledge of the fly genotype. Blinding was not performed during data analyses, because analyses were performed with the same automated algorithms and programming code. During analyses, wild-type control and mutant data sets are also easily identified and are directly compared one to another. |

# Reporting for specific materials, systems and methods

We require information from authors about some types of materials, experimental systems and methods used in many studies. Here, indicate whether each material, system or method listed is relevant to your study. If you are not sure if a list item applies to your research, read the appropriate section before selecting a response.

## Materials & experimental systems

| n/a | Involved in the study |
|---|---|
| ☐ | ☒ Antibodies |
| ☒ | ☐ Eukaryotic cell lines |
| ☒ | ☐ Palaeontology and archaeology |
| ☐ | ☒ Animals and other organisms |
| ☒ | ☐ Clinical data |
| ☒ | ☐ Dual use research of concern |

## Methods

| n/a | Involved in the study |
|---|---|
| ☒ | ☐ ChIP-seq |
| ☒ | ☐ Flow cytometry |
| ☒ | ☐ MRI-based neuroimaging |

# Antibodies

| | |
|---|---|
| Antibodies used | Mouse anti–α-Tubulin (clone 4.3; 1:3,000)(Walsh 1984) was obtained from the Developmental Studies Hybridoma Bank. The generation of polyclonal anti-Ste antibody (used at 1:10,000) was outsourced to Covance (Princeton, NJ) and was produced by immunizing guinea pigs with KLH conjugated Ac-KPVIDSSSGLLYGDEKKWC (53-70aa of Ste); horseradish peroxidase (HRP)-conjugated goat anti-mouse IgG (#115-035-003; 1:10,000; Jackson ImmunoResearch Laboratories), and anti-guinea pig IgG (#106-035-003; 1:10,000; Jackson ImmunoResearch Laboratories). |
| Validation | https://www.jacksonimmuno.com/catalog/products/106-035-003 ; https://www.jacksonimmuno.com/catalog/products/115-035-003 |

# Animals and other research organisms

Policy information about studies involving animals; ARRIVE guidelines recommended for reporting animal research, and Sex and Gender in Research

| | |
|---|---|
| Laboratory animals | Drosophila melanogaster w1118 (0–7 day old). The following lines were used: C(1)RM/C(X:Y)y1f1w1, armi1, armi72.1, aubHN2, aubQC42, zucEY11457, Df(2L)BSC323, nos-gal4:VP16, bam-gal4:VP16, UAS-flag3-myc6-ago3, UAS-gfp-aub, UAS-armi-gfp, UAS-dpp, TRIP.GL00254, TRIP.GL00076, TRIP.HMC02938, TRIP.HMS00373, TRIP.GL00111, UAS-gfp-Ste (SteXh:CG42398). |
| Wild animals | The study did not involve wild animals. |
| Reporting on sex | Findings specifically apply to male progeny from female X male crosses. |
| Field-collected samples | The study did not involve field-collected samples. |
| Ethics oversight | Work on Drosophila melanogaster does not require ethical oversight or experimental approval. |

Note that full information on the approval of the study protocol must also be provided in the manuscript.

