## [Peer Review File · Nature Cell Biology]

Peer Review Information

Journal: Nature Cell Biology

Manuscript Title: A maternally-programmed intergenerational mechanism enables male offspring to make piRNAs from Y-linked precursor RNAs in *Drosophila*

Corresponding author name(s): Professor Yukiko Yamashita

Editorial Notes:

Reviewer Comments & Decisions:

Decision Letter, initial version:
--

*Please delete the link to your author homepage if you wish to forward this email to co-authors.

Dear Dr Yamashita,

Thank you for submitting your manuscript, "A maternally-programmed intergenerational mechanism enables male offspring to utilize Y-linked piRNA in *Drosophila*", to Nature Cell Biology. I sincerely apologize for the delay in getting our decision to you. The manuscript has now been seen by 3 referees, who are experts in piRNAs, *Drosophila*, transposons (Referee #1); piRNAs, *Drosophila*, ncRNAs (Referee #2); and piRNAs, mouse (Referee #3). As you will see from their comments (attached below), they found the work of potential interest but raised substantial concerns, which in our view would need to be addressed with considerable revisions before we can consider publication in Nature Cell Biology.

Nature Cell Biology editors discuss the referee reports in detail within the editorial team, including the chief editor, to identify key referee points that should be addressed with priority, as opposed to requests that are beyond the scope of the current study. To guide the scope of the revisions, I have listed these points below. Our typical revision period is six months; we are committed to providing a fair and constructive peer-review process, so please feel free to contact me if you would like to discuss any of the referee comments further or anticipate any issues or delays addressing the reviews.

In particular, it would be essential to:

1- As per Rev#2 point #1, further addressing how the data fit with past work (ref 54) suggesting that the ping-pong machinery only may not drive the generation of Su(Ste) piRNAs will be important. Similarly, Reviewers #1 and #2 are not clear on the requirement for Aub or Ago3 for phased

biogenesis (Rev#1 points copied below; Rev#2 point #3, see also points #4, #5), and clarifying the mechanism experimentally should be a goal of the revision:

Rev#1: "- how do the authors explain the requirement for Ago3 in the process? Ago3 is not a recipient of phased piRNAs. Does this mean that the amplification of the hoppel trigger piRNA through ping-pong during embryonic/larval development is a requirement for the efficient triggering of the Su(Ste) transcript? Adding a comment here would be great.

- does the silencing of Stellate by Su(Ste) piRNAs generate Ago3-bound responder piRNAs from the Stellate transcripts in testes?"

2- All other reviewer comments about strengthening existing data, technical aspects, controls (including Rev#2 points #2, #7), methodological questions, and requests for clarifications, discussion, or text edits, should also be addressed.

3- Finally, please pay close attention to our guidelines on statistical and methodological reporting (listed below) as failure to do so may delay the reconsideration of the revised manuscript. In particular please provide:

We would be happy to consider a revised manuscript that would satisfactorily address these points, unless a similar paper is published elsewhere, or is accepted for publication in Nature Cell Biology in the meantime.

- ensure that it conforms to our format instructions and publication policies (see below and www.nature.com/nature/authors/).

- provide a point-by-point rebuttal to the full referee reports verbatim, as provided at the end of this letter.

- provide the completed Editorial Policy Checklist (found here <https://www.nature.com/authors/policies/Policy.pdf>), and Reporting Summary (found here <https://www.nature.com/authors/policies/ReportingSummary.pdf>). This is essential for reconsideration of the manuscript and these documents will be available to editors and referees in the event of peer review. For more information see <http://www.nature.com/authors/policies/availability.html> or contact me.

Nature Cell Biology is committed to improving transparency in authorship. As part of our efforts in this direction, we are now requesting that all authors identified as 'corresponding author' on published papers create and link their Open Researcher and Contributor Identifier (ORCID) with their account on the Manuscript Tracking System (MTS), prior to acceptance. ORCID helps the scientific community achieve unambiguous attribution of all scholarly contributions. You can create and link your ORCID from the home page of the MTS by clicking on 'Modify my Springer Nature account'. For more information please visit www.springernature.com/orcid.

[Redacted]

We hope that you will find our referees' comments and editorial guidance helpful. Please do not hesitate to contact me if there is anything you would like to discuss. Thank you again for considering the journal for your work, and again I sincerely apologize for the length of the process.

Best wishes,

Melina

Melina Casadio, PhD
Senior Editor, Nature Cell Biology
ORCID ID: <https://orcid.org/0000-0003-2389-2243>

Reviewers' Comments:

Reviewer #1:

Remarks to the Author:

In this manuscript, Venkei and colleagues demonstrate that the 5' to 3' directed phased piRNA biogenesis pathway allows utilisation of a Y-encoded transcript as piRNA biogenesis substrate through a maternally deposited transposon piRNA that cleaves the 5' end of the Y-linked transcript. This is a great paper that combines clever genetics with high end quantitative molecular biology.

I support publication after incorporation of the following comments:

- title: I am not sure whether this is the most intuitive title for the broad readership of this journal. Also: utilisation of a Y-linked piRNA is not really the point as it is utilisation of a Y-linked piRNA precursor, no?

- The authors write: "We detected expression of Su(Ste) piRNA precursor transcripts only from the genomic strand that produces transcripts antisense to Ste mRNAs (not shown)."
Please add the original data here.

- The authors write: "smRNA-FISH can detect Ste mRNAs and Su(Ste) precursor transcripts but not mature piRNAs, because small RNAs are not retained in formaldehyde-fixed tissues."
 Could the authors add a citation for this statement? How did Plasterk and colleagues detect microRNAs in zebrafish by in situ hybridisation?

- The authors write: "We conclude that maternal deposition of Su(Ste) piRNAs by XXY mothers suffices to silence Ste mRNA and bypasses the requirement for phased piRNA production pathway in early male germ cells."

This would mean that the maternally deposited piRNAs are stable until the onset of Stellate transcription, is this correct? Or do the authors envision that the maternally deposited piRNAs are capable of processing the Su(Ste) transcript through ping-pong? If the latter is the case, would this mean that the sense piRNAs of Stellate are essential? A clarifying comment would be helpful in the manuscript.

- how do the authors explain the requirement for Ago3 in the process? Ago3 is not a recipient of phased piRNAs. Does this mean that the amplification of the hoppel trigger piRNA through ping-pong during embryonic/larval development is a requirement for the efficient triggering of the Su(Ste) transcript? Adding a comment here would be great.

- does the silencing of Stellate by Su(Ste) piRNAs generate Ago3-bound responder piRNAs from the Stellate transcripts in testes?

- Figure 1:

I have a hard time to see where the f and g panels are related to the boxed areas in b and c.

- Figure 2: was the quantification of Su(Ste) quantification done in a z-stack so that each cell is properly analysed?

- Figure 5: please add a length scale for the 1360/Su(Ste) transcript.

- Figure 6a: why is the small RNA size profile cut at 23nt? Please include sizes down to 20nt.

- methods: Ct values were normalized to Gapdh by the $\Delta\Delta C_t$ method.
 Please add a citation

- please add molecular weight marker for western blots.

Finally, one maybe crazy idea: If one would express at high levels one or two siRNAs that are between the 1360 trigger piRNA target site and the Stellate sequence, then this might block the phasing process, thereby leading to the inability to silence Stellate. This is not an experiment that is required for this manuscript to be published, but it might be a neat test for the model.

Reviewer #2:

Remarks to the Author:

piRNAs are small non-coding RNAs that recognise complementary targets, such as transposons,

leading to their silencing. In *Drosophila*, piRNAs are maternally deposited through the oocyte. The male germline uses the piRNA pathway to repress *Stellate* (*Ste*), which, if active, leads to the production of *Ste* crystals and impairs male fertility. piRNAs capable of targeting *Ste* are produced from the *Suppressor of Stellate* [*Su(Ste)*] locus on the Y chromosome, and thus are not maternally deposited. How *Su(Ste)* piRNAs are generated in testes is currently not understood.

This manuscript by Venkei, Gainedinov et al. sheds light on the mechanism of *Su(Ste)* piRNA production in male germ cells. The authors used smRNA-FISH to visualise *Ste* and *Su(ste)* transcripts throughout *Drosophila* testis development and found that *Su(Ste)* is expressed earlier than its target *Ste*. Through temporally controlled knockdown and rescue experiments, they illustrate an early requirement of phased biogenesis components, and a late requirement of factors of the ping-pong amplification cycle.

The *Su(Ste)* promoter is localised within a 1360/Hoppel transposon insertion and as 1360/Hoppel piRNAs are maternally deposited, the authors hypothesised that cleavage by these piRNAs trigger phased piRNA biogenesis of the *Su(Ste)* transcript. To test this, they used XXY females expressing *Su(Ste)* from their Y chromosome. First, these females were able to produce piRNAs targeting *Su(Ste)* from the Y-linked *Su(Ste)* locus and pass these piRNAs onto their sons. Second, through sequencing of small and long RNAs from XXY ovaries, they found *Su(Ste)*-derived long 5'-monophosphorylated RNAs that displayed signatures consistent with anti-1360/Hoppel piRNA-guided cleavage and processing through the phased biogenesis pathway.

This work addresses the wider question on how intergenerational information is transferred through the piRNA pathway. Very little has been known about *Su(Ste)* piRNA production and this study contributes to understanding their biogenesis mechanism. The manuscript is well composed and easy to follow. However, several points as outlined below have to be addressed to fully support the authors' conclusions.

Major comments:

- 1) The ping-pong machinery has previously been reported as unlikely to drive the generation of *Su(Ste)* piRNAs (PMID 20980675) - this should be acknowledged. In the same study, one abundant piRNA dominated both the Aub- and Ago3-loaded piRNAs mapping to *Su(Ste)*, is this one also detected in the present study? If so, how does it relate to the phased biogenesis?
- 2) *Su(Ste)* has been described to produce sense transcripts giving rise to dsRNA. However, the authors did not detect any sense transcription, described as "data not shown". Given its importance, I think these data should be included and further expanded by stranded RNA-seq data. Similarly, I think it would be important to show both sense and antisense piRNAs (or the absence thereof) in Figure 6. Are any siRNAs detected?
- 3) Considering that Piwi seems not expressed in male germ cells, how can neither Aub or Ago3 be required for phased biogenesis (Extended Data Figure 2)? I don't follow how the *Su(Ste)*-derived piRNAs are retained to the later developmental stages if they are not loaded onto Aub or Ago3 during their biogenesis. If Aub/Ago3 act redundantly, can this be verified through a double-KD?
- 4) Fig 4 shows that *Ste* transcription is derepressed if Aub/Ago3 are knocked-down, while *Su(Ste)* mRNA is still processed (Extended Data Figure 2). What happens with the resulting *Su(Ste)* piRNAs?

Are they degraded after the cleavage as they are not loaded onto Aub or Ago3, what is the end results in Ste expression?

5) The rescue experiments (e.g., Figure 4g) are elegant. Similar could be done for Ago3. It may be interesting to also perform rescues in the context of Aub/Ago3 double-KDs.

6) I find the significance of the 5' monophosphorylated RNAs hard to interpret without putting the numbers in context. The authors claim that "For Su(Ste)-derived long RNAs overlapping both the upstream transposon insertion and the sequence complementary to Ste, the 5' ends of ~40% of these long RNAs lay between nucleotides g10 and g11 of an antisense maternal 1360/Hoppel piRNA". Is 40% more than what you would expect by chance? To know this, we need to check how many of the 1360/Hoppel positions that constitute piRNA 5' ends (threshold?). Also, is the 10-nt overlap stronger in this region compared to the downstream Su(Ste) region or other genomic regions? Is there a correlation between piRNA and cleavage product abundances? Is there a "ping-pong"-like signature? Please do present the data more exhaustive to aid interpretation.

7) Figure 5b: Is armi involved in degrading all 5' monophosphorylated long RNAs in the testis, or is this specific to the putative pre-pre-piRNAs in the 1360/Hoppel region? Please include suitable (negative and positive) controls.

8) Figure 7a: I have two major concerns with this analysis. First, I am not sure how relevant it is that Su(Ste) display phased biogenesis in ovaries. Any 5' monophosphorylated long RNA is likely to undergo phased biogenesis in the ovaries. However, we still do not know what happens in testis. Could the testis data discussed in Figure 5 be used instead? If not, I think the limitations of the current analysis should be highlighted. Second, I find it difficult to see that "5' ends of most Su(Ste) piRNAs in XXY ovaries concentrated in periodic peaks lying ~26 nt apart". Extended Figure 6a is more helpful but could be supplemented by checking for +1U signal, the hallmark of zuc-mediated cleavage. Phasing could be further quantified using cross correlation or Fourier transform. Is the phasing signature similar across both the 1360/Hoppel and Su(Ste) regions?

Minor comments:

1) Other factors, such as Spn-E, have been implicated in Ste suppression and this could be cited.

2) Extended Data Figure 1 and 2 have the same title.

3) Figure 2K: A t-test is not appropriate for discrete non-negative counts.

4) Row 161: "Figure 3a-c,g" should likely be "Figure 3a,c,g"? In general, the panel order could be reorganised here and elsewhere to make the figures easier to digest.

5) Please clarify what data is shown in Figure 7. How many cleavage sites are shown? How were they defined?

6) Please provide an overview of the high-throughput data generated and their key metrics.

7) The bioinformatic method description is very light and replicating some analyses (e.g., Figure 7), would be near impossible unless the authors make their scripts available or significantly increase the

amount of detail.

Reviewer #3:

Remarks to the Author:

This study elucidates the mechanism of the initial Y-linked piRNA biogenesis in *Drosophila* males, concerning the Ste-Su(Ste) piRNA-mediated suppression system. The smRNA-FISH analyses revealed that the Y-linked Su(ste) non-coding gene is expressed earlier than the X-linked Ste in male germ cells. A series of genetic experiments showed that Armi and Zuc are required for processing of Su(Ste) piRNA precursor transcripts in germline stem cells and spermatogonia. Most significantly, they showed that the maternal 1360/Hoppel-derived piRNAs initiated phased biogenesis of Su(Ste) piRNAs in males. This conclusion was further substantiated by studying XXY ovaries. The data are of high quality. The genetic experiments (knockdown, genetic mutation, and transgene) are elegant and informative. The conclusions are supported by the data. This study delineated the intergenerational mechanism for the maternal piRNA trigger that leads to silencing of Ste in male germ cells.

One minor comment on the last sentence in Discussion: "We speculate that this same mechanism may be used by mothers to protect their sons from selfish DNA in other species". This statement is too general. Do the authors mean other *Drosophila* species? It needs to be more specific or this speculation can be deleted.

Jeremy Wang

Methods should be written concisely, but should contain all elements necessary to allow interpretation and replication of the results. As a guideline, Methods sections typically do not exceed 3,000 words. The Methods should be divided into subsections listing reagents and techniques. When citing previous methods, accurate references should be provided and any alterations should be noted. Information must be provided about: antibody dilutions, company names, catalogue numbers and clone numbers

for monoclonal antibodies; sequences of RNAi and cDNA probes/primers or company names and catalogue numbers if reagents are commercial; cell line names, sources and information on cell line identity and authentication. Animal studies and experiments involving human subjects must be reported in detail, identifying the committees approving the protocols. For studies involving human subjects/samples, a statement must be included confirming that informed consent was obtained. Statistical analyses and information on the reproducibility of experimental results should be provided in a section titled "Statistics and Reproducibility".

All Nature Cell Biology manuscripts submitted on or after March 21 2016 must include a Data availability statement at the end of the Methods section. For Springer Nature policies on data availability see <http://www.nature.com/authors/policies/availability.html>; for more information on this particular policy see <http://www.nature.com/authors/policies/data/data-availability-statements-data-citations.pdf>. The Data availability statement should include:

- Accession codes for primary datasets (generated during the study under consideration and designated as "primary accessions") and secondary datasets (published datasets reanalysed during the study under consideration, designated as "referenced accessions"). For primary accessions data should be made public to coincide with publication of the manuscript. A list of data types for which submission to community-endorsed public repositories is mandated (including sequence, structure, microarray, deep sequencing data) can be found here <http://www.nature.com/authors/policies/availability.html#data>.
- Unique identifiers (accession codes, DOIs or other unique persistent identifier) and hyperlinks for datasets deposited in an approved repository, but for which data deposition is not mandated (see here for details <http://www.nature.com/sdata/data-policies/repositories>).
- At a minimum, please include a statement confirming that all relevant data are available from the authors, and/or are included with the manuscript (e.g. as source data or supplementary information), listing which data are included (e.g. by figure panels and data types) and mentioning any restrictions on availability.
- If a dataset has a Digital Object Identifier (DOI) as its unique identifier, we strongly encourage including this in the Reference list and citing the dataset in the Methods.

We recommend that you upload the step-by-step protocols used in this manuscript to the Protocol Exchange. More details can be found at www.nature.com/protocolexchange/about.

All imaging data should be accompanied by scale bars, which should be defined in the legend.

Cropped images of gels/blots are acceptable, but need to be accompanied by size markers, and to retain visible background signal within the linear range (i.e. should not be saturated). The boundaries of panels with low background have to be demarked with black lines. Splicing of panels should only be considered if unavoidable, and must be clearly marked on the figure, and noted in the legend with a statement on whether the samples were obtained and processed simultaneously. Quantitative comparisons between samples on different gels/blots are discouraged; if this is unavoidable, it should only be performed for samples derived from the same experiment with gels/blots were processed in parallel, which needs to be stated in the legend.

All placed images (i.e. a photo incorporated into a figure) should be on a separate layer and independent from any superimposed scale bars or text. Individual photographic images must be a

minimum of 300+ DPI (at actual size) or kept constant from the original picture acquisition and not decreased in resolution post image acquisition. All colour artwork should be RGB format.

The total number of Supplementary Figures (not including the “unprocessed scans” Supplementary Figure) should not exceed the number of main display items (figures and/or tables (see our Guide to Authors and March 2012 editorial <http://www.nature.com/ncb/authors/submit/index.html#suppinfo>; <http://www.nature.com/ncb/journal/v14/n3/index.html#ed>). No restrictions apply to Supplementary Tables or Videos, but we advise authors to be selective in including supplemental data.

GUIDELINES FOR EXPERIMENTAL AND STATISTICAL REPORTING

REPORTING REQUIREMENTS – To improve the quality of methods and statistics reporting in our papers we have recently revised the reporting checklist we introduced in 2013. We are now asking all life sciences authors to complete two items: an Editorial Policy Checklist (found here <https://www.nature.com/authors/policies/Policy.pdf>) that verifies compliance with all required editorial policies and a reporting summary (found here <https://www.nature.com/authors/policies/ReportingSummary.pdf>) that collects information on experimental design and reagents. These documents are available to referees to aid the evaluation of the manuscript. Please note that these forms are dynamic 'smart pdfs' and must therefore be downloaded and completed in Adobe Reader. We will then flatten them for ease of use by the reviewers. If you would like to reference the guidance text as you complete the template, please access these flattened versions at <http://www.nature.com/authors/policies/availability.html>.

Author Rebuttal to Initial comments
--

Venkei et al.

Nature Cell Biology submission NCB-A49585

Responses to Reviewers' Critiques

We thank the Reviewers for their useful comments, which helped us substantially improve the manuscript. Reviewers' recommendations inspired us to conduct additional experiments testing the proposed model of *Su(Ste)* piRNA biogenesis and function. In particular, we are grateful for the suggestion that, in addition to XXY ovaries, we examine in more detail the molecular mechanism of *Su(Ste)* piRNA biogenesis in testis. These new data offer additional evidence for phased processing of *Su(Ste)* precursors in early spermatogenesis.

Our point-by-point responses are below shown in blue text.

Editor:

In particular, it would be essential to:

1- As per Rev#2 point #1, further addressing how the data fit with past work (ref 54) suggesting that the ping-pong machinery only may not drive the generation of *Su(Ste)* piRNAs will be important.

Response: Data reported in Nagao et al., *RNA* 2010 indeed support the model proposed in our study. We revised the text to incorporate these earlier findings: e.g., the fact that most *Su(Ste)* piRNAs in Aub and Ago3 are derived from the antisense *Su(Ste)* precursors, and the requirements (1) for *armi* in *Su(Ste)* piRNA biogenesis and (2) for both Aub and Ago3 in *Ste* repression.

Similarly, Reviewers #1 and #2 are not clear on the requirement for Aub or Ago3 for phased biogenesis (Rev#1 points copied below; Rev#2 point #3, see also points #4, #5), and clarifying the mechanism experimentally should be a goal of the revision:

Response: The Editor's and the Reviewers' comments on the role of Aub and Ago3 in the proposed model encouraged us to conduct additional analyses and experiments (e.g., new data in Fig. 4) and to revise the text to better describe the role of the two proteins in *Su(Ste)* piRNA biogenesis and function.

Rev#1: "- how do the authors explain the requirement for Ago3 in the process? Ago3 is not a recipient of phased piRNAs. Does this mean that the amplification of the hoppel trigger piRNA through ping-pong during embryonic/larval development is a requirement for the efficient triggering of the *Su(Ste)* transcript? Adding a comment here would be great.

- does the silencing of *Stellate* by *Su(Ste)* piRNAs generate Ago3-bound responder piRNAs from the *Stellate* transcripts in testes?"

Response: Please see the detailed replies to the comments from Reviewer #1 and #2. Briefly, Nagao et al. and the data in this manuscript show that both Aub and Ago3 are

required for efficient silencing of *Ste* mRNAs. First, Nagao et al. noted that the majority of piRNAs in Aub and Ago3 are derived from antisense *Su(Ste)* precursor, i.e., both proteins are guided by piRNAs targeting *Ste* mRNAs. Second, in the absence of either Aub or Ago3, *Ste* mRNAs are derepressed (Nagao et al., *RNA* 2010, and Fig. 4 in this manuscript). These findings support the idea that both Aub and Ago3 are programmed with antisense *Su(Ste)* piRNAs during phased biogenesis in GSC/SGs. Aub and Ago3 therefore do not act redundantly, but additively: both proteins are required for efficient slicing of *Ste* mRNAs in spermatocytes. The revised manuscript also includes analyses (Supplementary Text 3) showing that the majority of *Ste*-derived piRNAs explained by *Su(Ste)*-guided cleavage are loaded in Ago3, not Aub. These data support the Reviewer's hypothesis that Ago3-loaded responder piRNAs are produced when *Ste* mRNAs are sliced by *Su(Ste)* piRNAs. However, these responder piRNAs cannot further propagate Ping-Pong piRNA amplification because *Su(Ste)* transcripts are not present in *Ste* expressing cells.

2- All other reviewer comments about strengthening existing data, technical aspects, controls (including Rev#2 points #2, #7), methodological questions, and requests for clarifications, discussion, or text edits, should also be addressed.

Response: We have added FISH data for the sense *Su(Ste)* precursor transcript and generated the requested stranded RNA-seq data (Rev#1 comment and Rev#2 point 2) in Extended Data Fig. 1. The controls for the analyses of 5' monophosphorylated long RNAs now appear in Extended Data Fig. 4 (Rev#2 point 7). The text was revised to include additional analyses, clarification, or references to discuss the detection of small RNAs in formaldehyde-fixed tissues, stability of small RNAs, sense and antisense piRNAs from both *Ste* and *Su(Ste)* loci and the absence of siRNA production from *Su(Ste)*, the probability of 10-nt overlap between piRNAs and 5' monophosphorylated long RNAs. We thank the Reviewers for pointing out ways to strengthen our analyses and clarify our interpretations.

3- Finally, please pay close attention to our guidelines on statistical and methodological reporting (listed below) as failure to do so may delay the reconsideration of the revised manuscript. In particular please provide:

Response: The revised manuscript adds a new Supplementary Figure 1 with uncropped Western blotting images from Fig. 3i and Extended Data Fig. 7.

Response: The new Supplementary Table 7 contains all numerical data from the main and Extended Data Figures.

Reviewer #1:

In this manuscript, Venkei and colleagues demonstrate that the 5' to 3' directed phased piRNA biogenesis pathway allows utilisation of a Y-encoded transcript as piRNA biogenesis substrate through a maternally deposited transposon piRNA that cleaves the 5' end of the Y-linked transcript. This is a great paper that combines clever genetics with high end quantitative molecular biology.

I support publication after incorporation of the following comments:

Response: We thank the Reviewer for their encouraging feedback. We have revised the manuscript to address the Reviewer's comments.

- title: I am not sure whether this is the most intuitive title for the broad readership of this journal. Also: utilisation of a Y-linked piRNA is not really the point as it is utilisation of a Y-linked piRNA precursor, no?

Response: The title was edited to "A maternally-programmed intergenerational mechanism enables male offspring to make piRNAs from Y-linked precursor RNAs in *Drosophila*".

- The authors write: "We detected expression of Su(Ste) piRNA precursor transcripts only from the genomic strand that produces transcripts antisense to Ste mRNAs (not shown)." Please add the original data here.

Response: Fluorescence in situ hybridization (FISH) data for both *sense* and *antisense* *Su(Ste)* transcripts now appear in the new Extended Data Figs. 1a, b. Consistent with the unlikely role of ping-pong amplification in the production of *Su(Ste)* piRNAs, only antisense *Su(Ste)* transcripts are detectable by FISH in germline stem cells and spermatogonia (GSC/SGs; Extended Data Fig. 1a). Few foci of sense *Su(Ste)* transcripts are detectable in spermatocytes, yet these foci appear several days after the transcription and processing of antisense *Su(Ste)* precursors occurs in GSC/SGs, suggesting that ping-pong between sense and antisense *Su(Ste)* precursors is unlikely. The requirement for the phased biogenesis pathway in earlier stages (GSC/SGs) also

argues against the idea that potential ping-pong in spermatocytes is functionally critical. These data concur with the strand-specific RNA-seq data that we added to the revised manuscript: in whole testes, steady-state abundance of sense *Su(Ste)* transcripts is $<1/10^{\text{th}}$ of the levels of antisense piRNA precursors (Extended Data Fig. 1c). We edited the text to include the discussion of these observations.

- The authors write: “smRNA-FISH can detect *Ste* mRNAs and *Su(Ste)* precursor transcripts but not mature piRNAs, because small RNAs are not retained in formaldehyde-fixed tissues.”

Could the authors add a citation for this statement? How did Plasterk and colleagues detect microRNAs in zebrafish by in situ hybridisation?

Response: The revised text adds the reference to Pena et al., *Nat Methods* 2009 reporting that “*in situ hybridization (ISH) using conventional formaldehyde fixation results in substantial microRNA loss from mouse tissue sections, which can be prevented by fixation with 1-ethyl-3-(3-dimethylaminopropyl) carbodiimide that irreversibly immobilizes the microRNA at its 5' phosphate.*” We speculate that Plasterk and colleagues detected the miRNA precursors—pre-miRNAs or pri-miRNAs—in their work.

- The authors write: “We conclude that maternal deposition of *Su(Ste)* piRNAs by XXY mothers suffices to silence *Ste* mRNA and bypasses the requirement for phased piRNA production pathway in early male germ cells.”

This would mean that the maternally deposited piRNAs are stable until the onset of *Stellate* transcription, is this correct? Or do the authors envision that the maternally deposited piRNAs are capable of processing the *Su(Ste)* transcript through ping-pong? If the latter is the case, would this mean that the sense piRNAs of *Stellate* are essential? A clarifying comment would be helpful in the manuscript.

Response: Argonaute-loaded small RNAs are one of the most stable RNA species in insect and mammalian cells. The median half-life of miRNAs in MEF cells is nearly 1.5 days (Kingston and Bartel, *Genome Res* 2019), and half-lives of all Ago2-loaded 2'-O-methylated small RNAs and many Ago1-loaded miRNAs in S2 cells are > 24 hours (Reichohlf et al., *Mol Cell* 2019). The remarkably slow turnover of small RNAs likely explains the efficient silencing of *Ste* mRNAs by the maternally deposited *Su(Ste)* piRNAs in sons of XXY mothers. In fact, the exceptional stability of Argonaute-protected small RNAs probably underlies the intergenerational inheritance of transposon-targeting piRNAs in animals with maternally deposited germplasm. We revised the text to add this discussion.

- how do the authors explain the requirement for Ago3 in the process? Ago3 is not a recipient of phased piRNAs. Does this mean that the amplification of the hoppe trigger piRNA through ping-

pong during embryonic/larval development is a requirement for the efficient triggering of the Su(Ste) transcript? Adding a comment here would be great.

Response: In testis and ovaries, transposon(TE)-derived piRNAs indeed partition between Aub and Ago3: most antisense, 1U-enriched TE piRNA are bound to Aub; most sense, 10A-biased TE piRNAs are loaded in Ago3 (Brennecke et al., *Cell* 2007; Nagao et al., *RNA* 2010).

By contrast, the majority of Su(Ste) piRNAs both in Ago3 and in Aub are antisense to *Ste* mRNAs and display a 1U-bias (as already shown by Nagao et al., *RNA* 2010; and new data in Extended Data Fig. 3e). Both Aub and Ago3 are required for *Ste* silencing (Nagao et al., *RNA* 2010; and Fig. 4 in this manuscript). These data support the idea that both Aub and Ago3 are programmed with antisense *Su(Ste)* piRNAs during phased processing of *Su(Ste)* precursors in GSC/SGs, and both Aub and Ago3 are required for efficient slicing of *Ste* mRNAs in spermatocytes.

- does the silencing of *Stellate* by Su(Ste) piRNAs generate Ago3-bound responder piRNAs from the *Stellate* transcripts in testes?

Response: As stated in the response to the Editor, the revised manuscript includes analyses (Supplementary Text 3) showing that the majority of *Ste*-derived piRNAs explained by *Su(Ste)*-guided cleavage are loaded into Ago3. These data support the Reviewer's hypothesis that Ago3-loaded responder piRNAs are produced when *Ste* mRNAs are sliced by *Su(Ste)* piRNAs.

- Figure 1: I have a hard time to see where the f and g panels are related to the boxed areas in b and c.

Response: For a more intuitive arrangement, the revised Fig. 1 now groups Z projections by genotype, not by cell type as in the original version. We also added the clarification in the legend. Also please note that the images of entire apical tip are Z-projections to better represent the overall signals, whereas images for each cell type are single Z plane (to avoid signals from multiple cells in depth). This is clarified in the revised legend.

- Figure 2: was the quantification of Su(Ste) quantification done in a z-stack so that each cell is properly analysed?

Response: The quantification is, in fact, based on the z-stack (max projection for signal intensity), which is reflected in the revised figure legend.

- Figure 5: please add a length scale for the 1360/Su(Ste) transcript.

Response: The length scale was added to Fig. 5.

- Figure 6a: why is the small RNA size profile cut at 23nt? Please include sizes down to 20nt.

Response: Small RNAs of 20–30-nt in length are now included in Figs. 1 and 6. The data in the revised Fig. 1 also attest to the lack of siRNA production from *Su(Ste)* transcripts, supporting the idea that only antisense *Su(Ste)* transcripts are expressed in early spermatogenesis.

- methods: Ct values were normalized to Gapdh by the $\Delta\Delta\text{Ct}$ method. Please add a citation

Response: The reference was added.

- please add molecular weight marker for western blots.

Response: The molecular weight was added to the revised Figures, and the uncropped images appear in Supplementary Figure 1.

Finally, one maybe crazy idea: If one would express at high levels one or two siRNAs that are between the 1360 trigger piRNA target site and the Stellate sequence, then this might block the phasing process, thereby leading to the inability to silence Stellate. This is not an experiment that is required for this manuscript to be published, but it might be a neat test for the model.

Response: We thank the reviewer for suggesting an interesting test for the model. siRNA-guided slicing of transcripts however generates 5' monophosphorylated cleavage products that are indistinguishable from RNA intermediates in the phased piRNA biogenesis (Han et al., *Science* 2015). siRNA-directed cleavage of piRNA precursor transcripts may thus result in increased phased piRNA production. In fact, siRNA slicing was recently proposed as an initiating event in de novo piRNA biogenesis in flies (Luo et al., *bioRxiv* 2022).

Reviewer #2:

Remarks to the Author:

piRNAs are small non-coding RNAs that recognise complementary targets, such as transposons, leading to their silencing. In *Drosophila*, piRNAs are maternally deposited through the oocyte. The male germline uses the piRNA pathway to repress Stellate (*Ste*), which, if active, leads to the production of *Ste* crystals and impairs male fertility. piRNAs capable of targeting *Ste* are produced from the Suppressor of Stellate [*Su(Ste)*] locus on the Y chromosome, and thus are not maternally deposited. How *Su(Ste)* piRNAs are generated in testes is currently not understood.

This manuscript by Venkei, Gainedinov et al. sheds light on the mechanism of Su(Ste) piRNA production in male germ cells. The authors used smRNA-FISH to visualise Ste and Su(ste) transcripts throughout *Drosophila* testis development and found that Su(Ste) is expressed earlier than its target Ste. Through temporally controlled knockdown and rescue experiments, they illustrate an early requirement of phased biogenesis components, and a late requirement of factors of the ping-pong amplification cycle.

The Su(Ste) promoter is localised within a 1360/Hoppel transposon insertion and as 1360/Hoppel piRNAs are maternally deposited, the authors hypothesised that cleavage by these piRNAs trigger phased piRNA biogenesis of the Su(Ste) transcript. To test this, they used XXY females expressing Su(Ste) from their Y chromosome. First, these females were able to produce piRNAs targeting Su(Ste) from the Y-linked Su(Ste) locus and pass these piRNAs onto their sons. Second, through sequencing of small and long RNAs from XXY ovaries, they found Su(Ste)-derived long 5'-monophosphorylated RNAs that displayed signatures consistent with anti-1360/Hoppel piRNA-guided cleavage and processing through the phased biogenesis pathway.

This work addresses the wider question on how intergenerational information is transferred through the piRNA pathway. Very little has been known about Su(Ste) piRNA production and this study contributes to understanding their biogenesis mechanism. The manuscript is well composed and easy to follow. However, several points as outlined below have to be addressed to fully support the authors' conclusions.

Response: We appreciate these encouraging comments. We have revised the manuscript as detailed below.

Major comments:

1) The ping-pong machinery has previously been reported as unlikely to drive the generation of Su(Ste) piRNAs (PMID 20980675) - this should be acknowledged.

Response: As mentioned in the response to the Editor, data reported in Nagao et al., *RNA* 2010 indeed support the model proposed in our study. We revised the text to incorporate these earlier findings: the fact that most *Su(Ste)* piRNAs in Aub and Ago3 are derived from the antisense *Su(Ste)* precursors, and the requirements (1) for *armi* in *Su(Ste)* piRNA biogenesis and (2) for both Aub and Ago3 in *Ste* repression.

In the same study, one abundant piRNA dominated both the Aub- and Ago3-loaded piRNAs mapping to *Su(Ste)*, is this one also detected in the present study? If so, how does it relate to the phased biogenesis?

Response: Consistent with processing of *Su(Ste)* precursors by Zuc, $77 \pm 1\%$ of *Su(Ste)*-derived piRNAs begin with a uridine and have no enrichment of adenine at

Response Figure 1: The complexity of data from Nagao et al., RNA 2010 and this manuscript. All datasets were down-sampled to 100,000 sequencing reads.

position 10 ($21 \pm 1\%$; new data in Extended Data Figure 3e). *Su(Ste)-4* (5'-UCUCAUCGUCGUAGAACAAGCCC[...]-3') is the most abundant *Su(Ste)*-derived piRNA both in our data ($\sim 1.5\%$ of all reads) and in data from Nagao et al. ($\sim 60\%$ of reads in Aub IP, $\sim 5\%$ of reads in Ago3 IP). The likely reason for the 3–40-fold difference in the fraction of *Su(Ste)-4* reads in our and the published data is the ligation bias during small RNA library preparation. The bias was first reported in Hafner et al., RNA 2011, after the work by Nagao et al. was published. Most current small RNA library preparation procedures—including the one used in this work—minimize the ligation bias by increasing ligase concentration, extending reaction time, using randomized portions in adapters, and conducting ligation in presence of PEG-8000 (e.g., see AQ-seq in Kim et al., NAR 2019). Consistent with this explanation, the complexity of datasets generated in this study is greater than for those from Nagao et al., even after down-sampling the data to the same sequencing depth (Response Fig. 1, below).

2) *Su(Ste)* has been described to produce sense transcripts giving rise to dsRNA. However, the authors did not detect any sense transcription, described as “data not shown”. Given its importance, I think these data should be included and further expanded by stranded RNA-seq data.

Response: Please see our response to a similar comment from Reviewer #1. Briefly, FISH data for the sense *Su(Ste)* transcripts now appear in the new Extended Data Figs. 1a, b. Consistent with the unlikely role of ping-pong amplification in the production of *Su(Ste)* piRNAs, only antisense *Su(Ste)* transcripts are detectable by FISH in germ stem cells and spermatogonia (GSC/SGs). Few foci of sense *Su(Ste)* transcripts are

detectable in spermatocytes, yet this observation agrees with the proposed model, as these foci appear several days after the transcription and processing of antisense *Su(Ste)* precursors commences in GSC/SGs. These data concur with the requested strand-specific RNA-seq data that we added to the revised manuscript: in whole testes, steady-state abundance of sense *Su(Ste)* transcripts is $<1/10^{\text{th}}$ of the levels of antisense piRNA precursors (Extended Data Fig. 1c).

Similarly, I think it would be important to show both sense and antisense piRNAs (or the absence thereof) in Figure 6. Are any siRNAs detected?

Response: The updated Figs. 1 and 6 now includes 20–30-nt small RNAs and demonstrates the lack of siRNA production from *Su(Ste)* transcripts, supporting the idea that only antisense *Su(Ste)* transcripts are expressed in early spermatogenesis. We also added the more detailed analyses of sense and antisense *Ste*- and *Su(Ste)*-derived piRNAs in Figs. 1 and 6 and Extended Data Fig. 8b.

3) Considering that Piwi seems not expressed in male germ cells, how can neither Aub or Ago3 be required for phased biogenesis (Extended Data Figure 2)? I don't follow how the *Su(Ste)*-derived piRNAs are retained to the later developmental stages if they are not loaded onto Aub or Ago3 during their biogenesis. If Aub/Ago3 act redundantly, can this be verified through a double-KD?

Response: As mentioned in the response to the Editor, the Reviewer's comment encouraged us to conduct additional analyses and experiments (see below) and revise the text to better describe the role of Aub and Ago3 in *Su(Ste)* piRNA biogenesis.

Data in Nagao et al. and in this manuscript show that both Aub and Ago3 are required for efficient silencing of *Ste* mRNAs. First, it had been already noted by the previous study that the majority of piRNAs in Aub and Ago3 are derived from antisense *Su(Ste)* precursor (Nagao et al., *RNA* 2010), i.e., both proteins are guided by piRNAs targeting *Ste* mRNAs. Second, in the absence of either Aub or Ago3, *Ste* mRNAs are derepressed (Nagao et al., *RNA* 2010, and Fig. 4 in this manuscript). These findings support the idea that both Aub and Ago3 are programmed with antisense *Su(Ste)* piRNAs during phased biogenesis in GSC/SGs. Aub and Ago3 therefore do not act redundantly, but additively: both proteins are required for efficient slicing of *Ste* mRNAs in spermatocytes.

In *bam*-driven rescue experiments, expression of transgenic Aub (Fig. 4g) and Ago3 (new data in Fig. 4i) starts at 4-cell SG stage, i.e., before the transcription of antisense *Su(Ste)* precursors reaches its peak. *Su(Ste)* piRNAs are thus loaded in Aub and Ago3 in early spermatogenesis and direct cleavage of *Ste* mRNAs later. We have updated the text and the figures accordingly (see new section "*Ste* Silencing Requires Expression of Both Aub and Ago3 in Spermatogonia").

On a separate note, although it is not relevant in the context of this revision, Piwi is indeed expressed in the male germline (Venkei et al., *PLoS Genet* 2020; <https://pubmed.ncbi.nlm.nih.gov/32168327/>). Piwi protein abundance in the germline is weaker than in the testicular soma, which is likely why it was missed in earlier studies, leading to the earlier notion that Piwi is not expressed in male germline.

4) Fig 4 shows that *Ste* transcription is derepressed if *Aub/Ago3* are knocked-down, while *Su(Ste)* mRNA is still processed (Extended Data Figure 2). What happens with the resulting *Su(Ste)* piRNAs? Are they degraded after the cleavage as they are not loaded onto *Aub* or *Ago3*, what is the end results in *Ste* expression?

Response: In *zuc* mutant and *armi* RNAi males, unprocessed *Su(Ste)* precursors are present in the cytoplasm of GSC/SGs, because endonucleolytic fragmentation of *Su(Ste)* transcripts is essentially blocked. Conversely, unprocessed *Su(Ste)* transcripts are not detectable when either *Aub* or *Ago3* are genetically removed or knocked down, because *Zuc* cleaves *Su(Ste)* precursors and piRNAs are likely loaded in *Aub* in *ago3^{RNAi}* males or in *Ago3* in *aub^{RNAi}* flies. Because both *Aub* and *Ago3* programmed with *Su(Ste)* piRNAs are required for efficient slicing of *Ste* mRNAs, presence of only *Aub* or only *Ago3* is not sufficient for *Ste* silencing. We have revised the text to clarify these data (Supplementary Text 2).

5) The rescue experiments (e.g., Figure 4g) are elegant. Similar could be done for *Ago3*. It may be interesting to also perform rescues in the context of *Aub/Ago3* double-KDs.

Response: The revised manuscript adds the data showing that *Ste* silencing can be restored in *ago3* mutant by *bam*-driven expression of transgenic *Ago3* (Fig. 4i). Because both *Aub* and *Ago3* are required for *Ste* silencing (Nagao et al., *RNA* 2010; and Fig. 4 in this work), the double knock-out experiment is unlikely to provide additional information.

6) I find the significance of the 5' monophosphorylated RNAs hard to interpret without putting the numbers in context. The authors claim that "For *Su(Ste)*-derived long RNAs overlapping both the upstream transposon insertion and the sequence complementary to *Ste*, the 5' ends of ~40% of these long RNAs lay between nucleotides g10 and g11 of an antisense maternal 1360/Hoppel piRNA". Is 40% more than what you would expect by chance? To know this, we need to check how many of the 1360/Hoppel positions that constitute piRNA 5' ends (threshold?). Also, is the 10-nt overlap stronger in this region compared to the downstream *Su(Ste)* region or other genomic regions? Is there a correlation between piRNA and cleavage product abundances? Is there a "ping-pong"-like signature? Please do present the data more exhaustive to aid interpretation.

Response: We thank the reviewer for their fair comment and useful suggestions. Additional analyses of our data showed that the 10-nt overlap between piRNAs and 5'

monophosphorylated long RNAs is, in fact, the most frequent occurrence compared to other arrangements ($Z_{10} = 8$, $p = 10^{-15}$; new data in Fig. 5b). The same pattern was observed in the piRNA-producing loci *42AB* ($Z_{10} = 5.1$, $p = 6 \times 10^{-7}$) and *petrel* ($Z_{10} = 8.5$, $p = 2.3 \times 10^{-17}$; new data in Extended Data Fig. 4a), but not in the *nos*, *bam*, and *bgcn* genic loci (new data in Extended Data Fig. 4a). The revised text adds these data.

7) Figure 5b: Is *armi* involved in degrading all 5' monophosphorylated long RNAs in the testis, or is this specific to the putative pre-pre-piRNAs in the 1360/Hoppel region? Please include suitable (negative and positive) controls.

Response: We included analyses of 5' monophosphorylated long RNAs derived from the piRNA producing loci *42AB* and *petrel* as positive controls, and from *nos*, *bam*, and *bgcn* genic loci as negative controls (new data in Extended Data Fig.4b). We also sequenced and analyzed 5' monophosphorylated long RNAs from *zuc* mutant males. Both *armi* RNAi and *zuc* mutant data show that 5' monophosphorylated RNAs only from piRNA-producing clusters *Su(Ste)*, *petrel*, and *42AB*—but not from genic loci—are stabilized when phased processing is blocked (new data in Extended Data Fig.4b). These results support the idea that 5' monophosphorylated long RNAs derived from piRNA producing loci are bona fide piRNA processing intermediates.

8) Figure 7a: I have two major concerns with this analysis. First, I am not sure how relevant it is that *Su(Ste)* display phased biogenesis in ovaries. Any 5' monophosphorylated long RNA is likely to undergo phased biogenesis in the ovaries. However, we still do not know what happens in testis. Could the testis data discussed in Figure 5 be used instead? If not, I think the limitations of the current analysis should be highlighted.

Response: The Reviewer's comment inspired us to conduct additional experiments aimed at understanding the biogenesis *Su(Ste)* piRNAs in testis. Initial analyses were hindered by the fact that an infinitesimally small fraction of the wild-type testis is the cell types in which *Su(Ste)* precursors are processed into piRNAs (i.e., GSC/SGs). To overcome this issue, we generated sequencing data from males with *nos*-driven expression of a transgenic copy of *decapentaplegic* (*dpp*; Xie and Spradling, *Cell* 1998). *Nos>dpp* males have the increased number of GSC/SGs, allowing examination of *Su(Ste)* piRNA biogenesis in more detail. The updated Figure 5 now contains data from testes. We thank the Reviewer for encouraging us to obtain more data to test our model.

Second, I find it difficult to see that "5' ends of most *Su(Ste)* piRNAs in XXY ovaries concentrated in periodic peaks lying ~26 nt apart". Extended Figure 6a is more helpful but could be supplemented by checking for +1U signal, the hallmark of *zuc*-mediated cleavage. Phasing could be further quantified using cross correlation or Fourier transform. Is the phasing signature similar across both the 1360/Hoppel and *Su(Ste)* regions?

Response: We note that, in the meta-analyses in the original Fig.7 and in the updated Fig. 5, we measure the distance of each *Su(Ste)* piRNA 5' end to the closest upstream 5' end of a monophosphorylated long RNA, i.e., the piRNA precursor. If precursors are fragmented into phased, tail-to-head piRNAs, the 5' ends of piRNAs are predicted to lie at regular intervals from the precursor 5' ends (Mohn et al., *Science* 2015; Han et al., *Science* 2015; Gainetdinov et al., *Mol Cell* 2018). In the updated Fig.5e, we added the autocorrelation analyses to test if *Su(Ste)* piRNA 5' ends occurred with consistent periodicity from precursor 5' ends. These analyses showed that *Su(Ste)* piRNA 5' ends lay in 25–26-nt intervals.

Unlike Piwi-loaded piRNAs, Aub- and Ago3-bound pre-piRNAs are trimmed by Nibbler, which often conceals their +1U bias (Hayashi et al., *Nature* 2016). Consistent with most *Su(Ste)* piRNAs loaded in Aub and Ago3, we failed to detect +1U bias for piRNAs from *Su(Ste)* loci. Nevertheless, in agreement with the endonucleolytic processing of *Su(Ste)* precursors by Zuc, we find that $77 \pm 1\%$ of *Su(Ste)*-derived piRNAs begin with a uridine and lack the enrichment of adenine at position 10 ($21 \pm 1\%$; new data in Extended Data Fig.3e)

Minor comments:

1) Other factors, such as Spn-E, have been implicated in Ste suppression and this could be cited.

Response: The reference was added in the Introduction.

2) Extended Data Figure 1 and 2 have the same title.

Response: We have corrected the title for Extended Data Fig. 2.

3) Figure 2K: A t-test is not appropriate for discrete non-negative counts.

Response: Unpaired, two-tailed Mann–Whitney U test (nonparametric, rank-order test for independent, ordinal groups) is used in the revised manuscript.

4) Row 161: “Figure 3a-c,g” should likely be “Figure 3a,c,g”? In general, the panel order could be reorganised here and elsewhere to make the figures easier to digest.

Response: We have reorganized panels in Figs. 2, 3, 4.

5) Please clarify what data is shown in Fig. 7. How many cleavage sites are shown? How were they defined?

Response: The updated legend for the figure states the number of cleavage sites used in the analyses.

6) Please provide an overview of the high-throughput data generated and their key metrics.

Response: The revised manuscript now includes Supplementary Table 6 that contains the requested information.

7) The bioinformatic method description is very light and replicating some analyses (e.g., Figure 7), would be near impossible unless the authors make their scripts available or significantly increase the amount of detail.

Response: We have deposited all the code used in this study at https://github.com/ildargv/Venkei_et_al_2023

Reviewer #3:

Remarks to the Author:

This study elucidates the mechanism of the initial Y-linked piRNA biogenesis in *Drosophila* males, concerning the Ste-Su(Ste) piRNA-mediated suppression system. The smRNA-FISH analyses revealed that the Y-linked Su(ste) non-coding gene is expressed earlier than the X-linked Ste in male germ cells. A series of genetic experiments showed that Armi and Zuc are required for processing of Su(Ste) piRNA precursor transcripts in germline stem cells and spermatogonia. Most significantly, they showed that the maternal 1360/Hoppel-derived piRNAs initiated phased biogenesis of Su(Ste) piRNAs in males. This conclusion was further substantiated by studying XXY ovaries. The data are of high quality. The genetic experiments (knockdown, genetic mutation, and transgene) are elegant and informative. The conclusions are supported by the data. This study delineated the intergenerational mechanism for the maternal piRNA trigger that leads to silencing of Ste in male germ cells.

One minor comment on the last sentence in Discussion: "We speculate that this same mechanism may be used by mothers to protect their sons from selfish DNA in other species". This statement is too general. Do the authors mean other *Drosophila* species? It needs to be more specific or this speculation can be deleted.

Jeremy Wang

Response: Thank you for the positive comments. The Discussion text now reads "Our study reveals a novel mechanism of intergenerational inheritance from mothers to sons, which may be utilized in other animal species that deposit germline determinants in oocytes."

Decision Letter, first revision:

Our ref: NCB-A49585A

14th June 2023

Dear Dr. Yamashita,

Please accept our apologies for the delay in sending a decision to you, as Reviewer#2 was unable to re-review your revisions. We secured Reviewer#1 to assess on your responses to Reviewer#2's previous concerns.

Thank you for submitting your revised manuscript "A maternally-programmed intergenerational mechanism enables male offspring to make piRNAs from Y-linked precursor RNAs in *Drosophila*" (NCB-A49585A). It has now been seen by the one of original referees, whom we have asked to comment on the responses to referee 2's points, and their comments are below. The reviewer finds that the paper has improved in revision, and therefore we'll be happy in principle to publish it in Nature Cell Biology, pending minor revisions to comply with our editorial and formatting guidelines.

We are now performing detailed checks on your paper and will send you a checklist detailing our editorial and formatting requirements in about a week.

**Please do not upload the final materials and make any revisions until you receive this additional information from us.

Please note that several figures need to be rearranged as they are in landscape format and this will not work with our layout. We would ask that you rearrange the panels of the figures which are in landscape format (main Figures 2, 3, 4, 6 and 7 and extended figures 3,4,5,6,7), to adhere to a maximum page size of roughly 180mm wide x 200mm high and use a font size of no smaller than 6pt Arial or Helvetica throughout for all figures (including main and ED figures). Also please use the full page space to fill each main figure, as each main figure should fill a full page.

Thank you again for your interest in Nature Cell Biology Please do not hesitate to contact me if you have any questions.

Best wishes,

Sabrya Carim, PhD
(she/her/hers)
Associate Editor, Nature Cell Biology
Nature Portfolio

Springer Nature
The Campus, 4 Crinan Street, London N1 9XW, UK
sabrya.carim@springernature.com
<https://orcid.org/0000-0001-9485-1938>

Reviewer #1 (Remarks to the Author):

The authors have done a great job in revising their work. They addressed the reviewer's comments, and the paper is ready for publication.

Decision Letter, final checks:

Our ref: NCB-A49585A

7th July 2023

Dear Dr. Yamashita,

Thank you for your patience as we've prepared the guidelines for final submission of your Nature Cell Biology manuscript, "A maternally-programmed intergenerational mechanism enables male offspring to make piRNAs from Y-linked precursor RNAs in *Drosophila*" (NCB-A49585A). Please carefully follow the step-by-step instructions provided in the attached file, and add a response in each row of the table to indicate the changes that you have made. Please also check and comment on any additional marked-up edits we have proposed within the text. Ensuring that each point is addressed will help to ensure that your revised manuscript can be swiftly handed over to our production team.

In recognition of the time and expertise our reviewers provide to Nature Cell Biology's editorial process, we would like to formally acknowledge their contribution to the external peer review of your manuscript entitled "A maternally-programmed intergenerational mechanism enables male offspring to make piRNAs from Y-linked precursor RNAs in *Drosophila*". For those reviewers who give their assent, we will be publishing their names alongside the published article.

Nature Cell Biology offers a Transparent Peer Review option for new original research manuscripts

submitted after December 1st, 2019. As part of this initiative, we encourage our authors to support increased transparency into the peer review process by agreeing to have the reviewer comments, author rebuttal letters, and editorial decision letters published as a Supplementary item. When you submit your final files please clearly state in your cover letter whether or not you would like to participate in this initiative. Please note that failure to state your preference will result in delays in accepting your manuscript for publication.

Cover suggestions

As you prepare your final files we encourage you to consider whether you have any images or illustrations that may be appropriate for use on the cover of Nature Cell Biology.

Nature Cell Biology has now transitioned to a unified Rights Collection system which will allow our Author Services team to quickly and easily collect the rights and permissions required to publish your work. Approximately 10 days after your paper is formally accepted, you will receive an email in providing you with a link to complete the grant of rights. If your paper is eligible for Open Access, our Author Services team will also be in touch regarding any additional information that may be required to arrange payment for your article.

Please note that *Nature Cell Biology* is a Transformative Journal (TJ). Authors may publish their research with us through the traditional subscription access route or make their paper immediately open access through payment of an article-processing charge (APC). Authors will not be required to make a final decision about access to their article until it has been accepted. Find out more about Transformative Journals

Please note that you will not receive your proofs until the publishing agreement has been received

through our system.

[Redacted]

Best regards,

Kendra Donahue
Staff
Nature Cell Biology

On behalf of

Sabrya Carim, PhD
(she/her/hers)
Associate Editor, Nature Cell Biology
Nature Portfolio

Springer Nature
The Campus, 4 Crinan Street, London N1 9XW, UK
sabrya.carim@springernature.com
<https://orcid.org/0000-0001-9485-1938>

Reviewer #1:

Remarks to the Author:

The authors have done a great job in revising their work. They addressed the reviewer's comments, and the paper is ready for publication.

[Redacted]

Final Decision Letter:

Dear Dr Yamashita,

I am pleased to inform you that your manuscript, "A maternally-programmed intergenerational mechanism enables male offspring to make piRNAs from Y-linked precursor RNAs in *Drosophila*", has now been accepted for publication in *Nature Cell Biology*.

Over the next few weeks, your paper will be copyedited to ensure that it conforms to *Nature Cell Biology* style. Once your paper is typeset, you will receive an email with a link to choose the appropriate publishing options for your paper and our Author Services team will be in touch regarding any additional information that may be required.

Publication is conditional on the manuscript not being published elsewhere and on there being no announcement of this work to any media outlet until the online publication date in *Nature Cell Biology*.

Please note that *Nature Cell Biology* is a Transformative Journal (TJ). Authors may publish their research with us through the traditional subscription access route or make their paper immediately open access through payment of an article-processing charge (APC). Authors will not be required to make a final decision about access to their article until it has been accepted. Find out more about Transformative Journals

Authors may need to take specific actions to achieve compliance with funder and institutional open access mandates. If your research is supported by a funder that requires immediate open access (e.g. according to Plan S principles) then you should select the gold OA route, and we will direct you to the compliant route where possible. For authors selecting the subscription

publication route, the journal's standard licensing terms will need to be accepted, including self-archiving policies. Those licensing terms will supersede any other terms that the author or any third party may assert apply to any version of the manuscript.

If you have not already done so, we strongly recommend that you upload the step-by-step protocols used in this manuscript to the Protocol Exchange (www.nature.com/protocolexchange), an open online resource established by Nature Protocols that allows researchers to share their detailed experimental know-how. All uploaded protocols are made freely available, assigned DOIs for ease of citation and are fully searchable through nature.com. Protocols and Nature Portfolio journal papers in which they are used can be linked to one another, and this link is clearly and prominently visible in the online versions of both papers. Authors who performed the specific experiments can act as primary authors for the Protocol as they will be best placed to share the methodology details, but the Corresponding Author of the present research paper should be included as one of the authors. By uploading your Protocols to Protocol Exchange, you are enabling researchers to more readily reproduce or adapt the methodology you use, as well as increasing the visibility of your protocols and papers. You can also establish a dedicated page to collect your lab Protocols. Further information can be found at www.nature.com/protocolexchange/about

With kind regards,

Sabrya Carim, PhD
(she/her/hers)
Associate Editor, Nature Cell Biology
Nature Portfolio

Springer Nature
The Campus, 4 Crinan Street, London N1 9XW, UK
sabrya.carim@springernature.com
<https://orcid.org/0000-0001-9485-1938>